# Hierarchically Gated Recurrent Neural Network for Sequence Modeling

[1]**Zhen Qin**[*],   [2]**Songlin Yang**[*],   [1]**Yiran Zhong**[✉]
[1]OpenNLPLab, Shanghai Artificial Intelligence Laboratory, [2]MIT CSAIL
`https://github.com/OpenNLPLab/HGRN`

## Abstract

Transformers have surpassed RNNs in popularity due to their superior abilities in parallel training and long-term dependency modeling. Recently, there has been a renewed interest in using linear RNNs for efficient sequence modeling. These linear RNNs often employ gating mechanisms in the output of the linear recurrence layer while ignoring the significance of using forget gates within the recurrence. In this paper, we propose a gated linear RNN model dubbed Hierarchically Gated Recurrent Neural Network (HGRN), which includes forget gates that are lower bounded by a learnable value. The lower bound increases monotonically when moving up layers. This allows the upper layers to model long-term dependencies and the lower layers to model more local, short-term dependencies. Experiments on language modeling, image classification, and long-range arena benchmarks showcase the efficiency and effectiveness of our proposed model. The source code is available at https://github.com/OpenNLPLab/HGRN.

## 1  Introduction

Sequence modeling is a fundamental problem in various domains such as natural language processing [12, 43, 44, 61, 64], time series analysis [84], computer vision [3, 13, 45, 74], and audio processing [1, 18, 73]. Prior to the invention of Transformers [81], RNN and its variants were the primary selections of architectures for sequence modeling, and have been widely used in machine translation [6], stock price prediction [68], weather forecasting [65], speech recognition [51], and *etc*.

RNNs have two main drawbacks: slow sequential training and limited capability in modeling long-term dependencies. With the swift development of deep learning and the pervasive use of GPUs, these drawbacks prevent it from flourishing in modern long-sequence modeling tasks. Meanwhile, Transformers [81] have rapidly gained popularity and now dominate various research areas in sequence modeling due to their better abilities in parallel training and long-term dependency modeling. However, Transformer's quadratic time complexity makes long sequence modeling expensive. On the other hand, RNN offers linear complexity and serves as an ideal choice for long sequence modeling. This works aim to addressing these RNN drawbacks, revitalizing their applicability in long-sequence modeling tasks.

To address the training inefficiency problem, we turn to more efficient RNN variants that employ element-wise linear recurrence (ELR) relations [48]. ELR provides two main advantages: (i) By removing nonlinearities in the recurrence, it enables parallelized training. (ii) By assuming independence between distinct hidden states, it enables efficient hidden state updates (through element-wise product instead of matrix multiplication) [20, 40]. Notably, ELR has been used in many modern linear RNN models, including the diagonalized versions of structured state-space models [21] (S4) [20, 26, 71] and RWKV [55]. In recent advancements, numerous studies have incorporated

---

[*]Equal contribution. [✉] Indicates corresponding author (Email address: *zhongyiran@gmail.com*).

37th Conference on Neural Information Processing Systems (NeurIPS 2023).

gating mechanisms into the outputs of linear recurrence layers [11, 46, 49, 55, 82], similar to the output gates in LSTMs and leading to considerable performance gains. However, most current studies overlook the significance of the forget gate, which is often regarded as the most important gate in LSTMs [19, 80]. In this work, we underscore the importance of employing forget gates in linear RNNs and adopt gated linear RNNs for both efficiency and high performance.

To effectively capture long-term dependencies in gated RNNs, it is crucial to maintain high forget gate values close to one [23]. However, gates in saturated regimes (i.e., close to zero or one) suffer from the gradient vanishing issue [23]. Moreover, if all forget gate values are close to one, RNNs will not be able to effectively forget irrelevant information, compromising their ability to model short-term dependencies. To address these challenges, we introduce Hierarchically Gated Recurrent Units(**HGRU**). In **HGRU**, we add an additive learnable value, referred to as the lower bound, to the original forget gate value, effectively mitigating the issue of saturated gates [23] by pushing gate activations away from the saturated regimes. Furthermore, we design the lower bounds to increase monotonically as we move up the layers of the RNN. This ensures that the forget gate values in the lower layers remain relatively small, enabling the necessary forgetting of past information for modeling short-term dependencies. In contrast, in the uppermost layer, the forget gate values approach one, facilitating the effective modeling of long-term dependencies. Our proposed model has proven to be highly efficient and effective, as demonstrated by its outstanding performance in language modeling, image classification, and long-range arena benchmarks.

## 2 Related work

**Efficient token mixing for sequence modeling.** [83] abstracts self-attention (SA) as token mixing, thereby transforming the Transformer architecture into MetaFormer. MetaFormer comprises essential components such as token mixer, channel mixer, residual connections, and LayerNorm. This abstraction highlights that the success of Transformers does not solely rely on SA but rather on the holistic integration of these components. Notably, token mixers can be replaced with simpler alternatives like pooling layers without compromising the model's performance in the context of vision transformer. For sequence modeling tasks, [29] provides a comprehensive analysis and discussion of different token mixing strategies. Two prominent contenders, long convolution and linear recurrence, show promise as replacements for SA modules in long sequence modeling due to their superior asymptotic time complexity and competitive performances. In long convolution models [14, 41, 57, 59], the kernel size matches the input sequence length, enabling a broader context compared to traditional convolutions. Training is accomplished using the efficient $\mathcal{O}(n \log n)$ fast Fourier transforms (FFT) algorithm. However, long convolutions face challenges such as the need for causal convolution inference, which requires caching all historical computations similar to the key-value (KV) cache in SA. This can lead to memory limitations when processing long sequences. Moreover, the inference complexity of long convolutions remains higher than that of RNNs. These factors make linear RNNs a more suitable alternative to replace SA in long-sequence modeling. TransNormerLLM [61] scales efficient token mixing in large language models to achieve competitive performance and superior training and inference efficiency compared to transformer-based models.

**Element-wise linear recurrence.** The slower training speeds of traditional RNNs can be attributed to two main factors: (i) The updating of the hidden state involves full matrix multiplication. (ii) The presence of nonlinearity within the recurrence prevents parallel computation. To tackle the first issue, [40] introduced a simplified interaction between hidden states. This allowed the hidden state update to be performed using an element-wise product instead of full matrix multiplication. They demonstrated that this approach is notably fast when the (nonlinear) recurrence for each dimension is fused within a single CUDA kernel. Likewise, for the linear case, diagonalized versions of S4 [20, 26] have also exhibited speed improvements over S4 by leveraging element-wise recurrence. Regarding the second challenge, the ability to capture nonlinear dependencies on past data can be achieved by stacking multiple linear recurrence layers interleaved with nonlinear MLP blocks. This indicates the potential to eliminate nonlinearity, as suggested by [4, 25, 48]. Empirical support for this strategy's effectiveness came later, as demonstrated by [11, 20, 24, 53, 55, 71]. [52] further highlighted that such an architecture still possesses Universal Approximator properties, thus justifying the employment of linear recurrence. By excluding nonlinearity, [48, 71] showed that the parallel scan algorithm can be used for parallel training.

Linear recurrence can be broadly categorized into exponential moving averages (EMA) and gating schemes, as noted by [48]. The key difference is whether the decay rate is data-dependent. Models such as S4 [21], S4D [20], MEGA [46], RWKV [55], and LRU [53] utilize the EMA approach, where the decay rate is static for all time steps (i.e., data-independent), while our model uses a data-dependent dynamic decay rate through the use of the forget gate. We remark on the importance of incorporating a data-dependent decay rate, which is largely ignored by current works in linear RNNs. Although liquid S4 [28] uses a dynamic transition matrix (which amounts to a data-dependent decay rate), it employs a limited form for FFT-based training. Our model does not have the convolutional view and thus cannot use FFT for training but allows the use of parallel scan.

The field of linear Transformers and linear RNNs exhibits a close relationship. [34] shows that linear Transformers can be reformulated as RNNs during auto-regressive decoding, revealing similarities to the update rules observed in fast weight additive outer products [66, 67]. These updates can be seen as a special case of element-wise linear recurrence, where forget gate values are consistently set to one across time and hidden states are two-dimensional. However, this formulation in linear Transformers lacks the ability to forget irrelevant information, resulting in the attention dilution issue [60]. To address this limitation, [66] introduced the delta rule to forget values associated with the current write key by removing the corresponding value before adding the new value. Alternatively, [47, 56] proposed gating mechanisms similar to those in gated RNNs to facilitate the forgetting of irrelevant information.

**Long-term dependencies in RNNs.** RNNs fall short in long-term dependency modeling, which is commonly attributed to the gradient vanishing issue. Three methods are typically applied to mitigate this issue. (i) Gating mechanisms [9, 17, 23, 30, 70], which are believed to be crucial to the success of LSTMs, use additive (instead of multiplicative) hidden state update rules to improve gradient flow. (ii) Regularizing or initializing the eigenvalues of the recurrent weight matrix (close) to one via identity matrices [38] or unitary matrices [2]. In the diagonal linear RNN case, the eigenvalues coincide with the element-wise decay rates, and LRU [53] uses randomized linear algebra techniques to initialize eigenvalues to be close to one. [53] also interestingly points out that many modern state-space models use a very small time step value on initialization for discretization, resulting in eigenvalues or decay rates close to one. (iii) Adding skip connections between distant time steps to allow shortcuts for gradient flow [5, 8, 37]. Our approach combines (i) and (ii), which improves gating mechanisms with a regularized dynamic decay rate that approaches one in the upper layer.

## 3 Method

### 3.1 Architecture overview

Our proposed Hierarchically Gated Recurrent Network (**HGRN**) is depicted in Figure 1. It has multiple stacked layers, each of which consists of a token mixing module **HGRU** and a channel mixing module **GLU** (Gated Linear Unit [69]).

### 3.2 HGRU exploration

We begin with a simple gated linear recurrent layer, which is defined as:

$$\mathbf{f}_t = \text{Sigmoid}\left(\mathbf{x}_t \mathbf{W}_f + \mathbf{b}_f\right) \in \mathbb{R}^{1 \times d},$$
$$\mathbf{i}_t = \text{Sigmoid}\left(\mathbf{x}_t \mathbf{W}_i + \mathbf{b}_i\right) \in \mathbb{R}^{1 \times d},$$
$$\mathbf{c}_t = \text{SiLU}\left(\mathbf{x}_t \mathbf{W}_t + \mathbf{b}_z\right) \in \mathbb{R}^{1 \times d}, \quad (1)$$
$$\mathbf{h}_t = \mathbf{f}_t \odot \mathbf{h}_{t-1} + \mathbf{i}_t \odot \mathbf{c}_t \in \mathbb{R}^{1 \times d},$$
$$\mathbf{h}_0 = \mathbf{0} \in \mathbb{R}^{1 \times d},$$

where $\odot$ denotes the element-wise product. Following the terminology used in the RNN literature, we refer to $\mathbf{f}_t$ and $\mathbf{i}_t$ as the forget and input gates, respectively. It is worth noting that $\mathbf{f}_t$ and $\mathbf{i}_t$ depend only on $\mathbf{x}_t$ and not on $\mathbf{h}_{t-1}$. This characteristic enables the use of the parallel scan algorithm [48, 71], otherwise it is infeasible. We then make the following changes toward our final **HGRU** step by step.

**Algorithm 1** Recurrent Computing

1: Input: $\mathbf{c}_t \in \mathbb{C}^{1 \times d}, \mu_t, \theta, \gamma^k \in \mathbb{R}^{1 \times d}, t = 1, \ldots, n, k = 1, \ldots, H.$
2: Init: $\mathbf{h} = \mathbf{0} \in \mathbb{C}^{1 \times d}, \mathbf{H} \in \mathbb{C}^{n \times d}.$
3: **for** $t = 1$ **to** $n$ **do**
4:     **begin**
5:     $\lambda_t = \gamma^k + (1 - \gamma^k) \odot \mu_t.$
6:     $\mathbf{h} = \lambda_t \exp(i\theta)\mathbf{h} + (1 - \lambda_t)\mathbf{c}_t.$
7:     $[\mathbf{H}]_t = \mathbf{h}.$
8:     **end**
9: return $\mathbf{H}.$

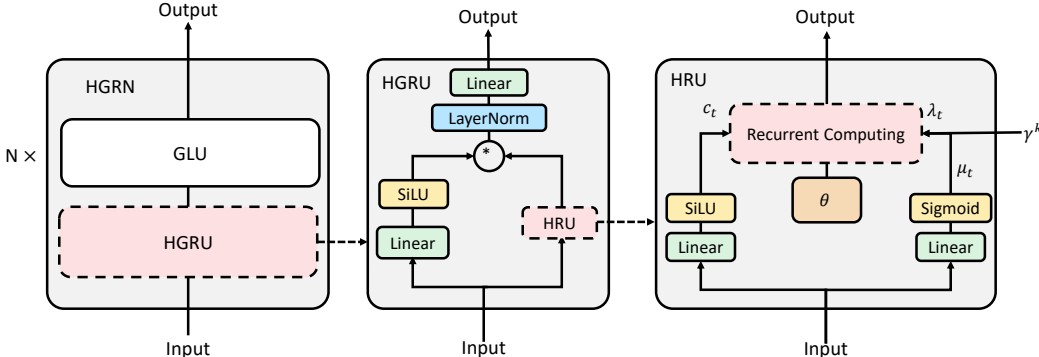

Figure 1: Illustration of the neural architecture. Each **HGRN** layer consists of a token mixer **HGRU** and a channel mixer GLU. **HGRU** employs linear recurrence in the complex domain: $\mathbf{h}_t = \lambda_t \odot \exp(i\theta) \odot \mathbf{h}_{t-1} + (1 - \lambda_t) \odot \mathbf{c}_t$. Here $c_t$ is the input vector, $\theta$ is the rotation angle, $\mu_t$ is the output of the original forget gate, $\gamma^k$ is the lower bound of the $k$th layer, $\lambda$ is the resulting data dependent decay rate: $\lambda_t = \gamma^k + (1 - \gamma^k) \odot \mu_t$.

**Complex-valued recurrence.** For linear RNNs with static decay rates, it is common to perform eigendecompositions on the recurrent weight matrix to achieve element-wise linear recurrence. However, if only real-valued eigenvalues are allowed, it restricts the range of the recurrent weight matrix to be symmetric, limiting the model's expressiveness. To overcome this limitation, linear RNNs often employ complex-valued eigenvalues to enhance the model's expressive power [20, 26, 27, 32, 53]. Motivated by this, we extend our model to consider $\mathbf{h}_t, \mathbf{i}_t, \mathbf{c}_t \in \mathbb{C}^{1 \times d}$ as complex values. For the input $c_t$, we parameterize its real and imaginary parts separately as follows:

$$\mathrm{Re}(\mathbf{c}_t) = \mathrm{SiLU}\left(\mathbf{x}_t \mathbf{W}_{cr} + \mathbf{b}_{cr}\right) \in \mathbb{R}^{1 \times d},$$
$$\mathrm{Im}(\mathbf{c}_t) = \mathrm{SiLU}\left(\mathbf{x}_t \mathbf{W}_{ci} + \mathbf{b}_{ci}\right) \in \mathbb{R}^{1 \times d}.$$

Regarding the forget gate values, we find it convenient to use the exponential representation of complex numbers and parameterize $\mathbf{f}_t$ as follows: $\mathbf{f}_t = \lambda_t \odot \exp(i\theta_t)$. Here, $i^2 = -1$, $\lambda_t, \theta_t \in \mathbb{R}^d$ and $\exp(i\theta_t) = \cos\theta_t + \sin\theta_t i$. The magnitude argument $\lambda_t$ determines the intensity of remembering historical information, while the phase argument $\theta_t$ determines the oscillation frequencies. We find that parameterizing $\theta_t$ in a data-independent manner is preferable, as it allows for a clear interpretation of encoding relative position information (see next subsection for more discussions) , which is reminiscent of Rotary Positional Embedding (RoPE) [72]. We shared $\theta$ arcoss times steps, i.e., $\mathbf{f}_t = \lambda_t \odot \exp(i\theta)$, initialize $\theta$ as RoPE does, but make it learnable like LRPE [63].

**Lower bound on forget gate values.** Since the intensity of remembering information is only related to the magnitude argument $\lambda_t$, we focus on how to add a lower bound to $\lambda_t$. As mentioned earlier, we want to set a monotonically increasing lower bound on the forget gate (magnitude) values. Inspired by ON-LSTM [70], we employ the `cummax` activation function to achieve this. Concretely, we allocate $\mathbf{\Gamma} \in \mathbb{R}^{H \times d}$ to parameterize lower bounds independently for all hidden states, where $H$ is the number of layer. Assuming the layer index is $k$, we have the following calculations:

$$\mathbf{P} = (\mathrm{Softmax}(\mathbf{\Gamma}, \dim = 0) \in \mathbb{R}^{H \times d},$$
$$\gamma^k = [\mathrm{Cumsum}(\mathbf{P}, \dim = 0)]_k \in \mathbb{R}^{1 \times d}.$$

Here we define $[\mathrm{Cumsum}(\mathbf{x})]_k = \left(\sum_{i=1}^{k} x_i\right) - x_1$ to prevent the highest layer's lower bound from being one as we still want the ability to forget irrelevant information.

We remark that there is a difference in the use of `cummax` between our model and ON-LSTM. In ON-LSTM, `cummax` is applied to the hidden state dimension within a single layer, while in our case, we apply `cummax` on the layer dimension across different layers to enable upper layers to model long-range dependencies.

Finally, $\lambda_t$ in the $k$-th layer is parameterized as follows:

$$\mu_t = \text{Sigmoid}\left(\mathbf{x}_t \mathbf{W}_\mu + \mathbf{b}_\mu\right) \in \mathbb{R}^{1 \times d},$$
$$\lambda_t = \gamma^k + (1 - \gamma^k) \odot \mu_t \in \mathbb{R}^{1 \times d}.$$

Comparing to before (i.e., without lower bounds), to achieve the same forget rate value $\bar{\gamma}$ closed to one, $\mu_t$ will be pushed away from the Sigmoid activation function's saturated regions (i.e., near one),

$$\mu_t = \frac{\bar{\gamma} - \gamma^k}{1 - \gamma^k} < \bar{\gamma},$$

thereby mitigating the gradient vanishing issue [23] and making gradient-based optimization easier.

**Tying input and forget gates.** To reduce the number of parameters, it is common to use leaky units, i.e., tying the input gate with the forget gate using $\mathbf{i}_t = 1 - \mathbf{f}_t$, which has a close relationship to the discretization of continuous-time system [75] and exponential moving average [33], and has been proven effective empirically [9, 19]. To allows for a clear interpretation of encoding relative position information, we only apply this strategy on the magnitude argument:

$$\mathbf{h}_t = \lambda_t \odot \exp(i\theta) \odot \mathbf{h}_{t-1} + (1 - \lambda_t) \odot \mathbf{c}_t \in \mathbb{C}^{1 \times d}. \tag{2}$$

**Output gates and projection.** The addition of gates to the output of the recurrence layer has been shown to be effective in state-space models [11, 46, 49, 82]. Motivated by this, we incorporate an output gate before performing the output projection as follows and get **HGRU**:

$$\mathbf{g}_t = \text{Sigmoid}(W_g \mathbf{x}_t + b_g) \in \mathbb{R}^{1 \times 2d},$$
$$\mathbf{o}_t' = \text{LayerNorm}(\mathbf{g}_t \odot [\text{Re}(\mathbf{h}_t), \text{Im}(\mathbf{h}_t)]) \in \mathbb{R}^{1 \times 2d}, \tag{3}$$
$$\mathbf{o}_t = \mathbf{o}_t' \mathbf{W}_o + \mathbf{b}_o \in \mathbb{R}^{1 \times d}.$$

### 3.3 Token mixing perspective of HGRU

We provide the token mixing perspective of **HGRU** similar to [32]. Expanding Equation 2, we have:

$$\mathbf{h}_t = \sum_{s=1}^{t} (1 - \lambda_s) \left[ \prod_{k=s+1}^{t} \lambda_k \exp(i\theta) \right] \mathbf{c}_s = \sum_{s=1}^{t} (1 - \lambda_s) \left[ \prod_{k=s+1}^{t} \lambda_k \right] \exp(i(t-s)\theta) \mathbf{c}_s \tag{4}$$

Written in matrix form, we have:

$$\mathbf{H} = \begin{bmatrix} \mathbf{h}_1 \\ \vdots \\ \vdots \\ \mathbf{h}_n \end{bmatrix}, \mathbf{A} = \begin{bmatrix} 1 - \lambda_1 & 0 & \cdots & 0 \\ (1-\lambda_1)\lambda_2 \exp(i\theta) & 1 - \lambda_2 & & \vdots \\ \vdots & \vdots & \ddots & 0 \\ (1-\lambda_1)\left[\prod_{s=2}^{n} \lambda_k\right]\exp(i(n-1)\theta) & \cdots & \cdots & 1 - \lambda_n \end{bmatrix}, \mathbf{C} = \begin{bmatrix} \mathbf{c}_1 \\ \vdots \\ \vdots \\ \mathbf{c}_n \end{bmatrix} \tag{5}$$

So the token mixing module can be formed as follows:

$$\mathbf{H} = \mathbf{A}\mathbf{C}. \tag{6}$$

Note that the token mixing matrix $\mathbf{A}$ can be decomposed into two parts $\mathbf{A} = \mathbf{\Lambda} \odot \mathbf{\Theta}$:

$$\mathbf{\Lambda} = \begin{bmatrix} 1 - \lambda_1 & 0 & \cdots & 0 \\ (1-\lambda_1)\lambda_2 & 1 - \lambda_2 & & \vdots \\ \vdots & \vdots & \ddots & 0 \\ (1-\lambda_1)\left[\prod_{s=2}^{n} \lambda_k\right] & \cdots & \cdots & 1 - \lambda_n \end{bmatrix}, \mathbf{\Theta} = \begin{bmatrix} 1 & 0 & \cdots & 0 \\ \exp(i\theta) & 1 & & \vdots \\ \vdots & \vdots & \ddots & 0 \\ \exp(i(n-1)\theta) & \cdots & \cdots & 1 \end{bmatrix} \tag{7}$$

This decomposition means that the Token mixing matrix $\mathbf{\Lambda}$ can be decoupled into two independent modules, where $\mathbf{\Lambda}$ models the long-distance dependency and $\mathbf{\Theta}$, a Toeplitz matrix, models the relative positional relationship and enhanced expressiveness. Note that if $\mathbf{\Theta}$ depends on the input, then the matrix $\mathbf{\Lambda}$ will no longer be a Toeplitz matrix, thus unable to capture relative position information. It can be also viewed as a RoPE-enhanced attention mechanism: $\mathbf{\Lambda}$ corresponds to the attention matrix but the attention score here is the cumulative product of data-dependent decay rates; $\mathbf{\Theta}$ directly corresponds to RoPE.

Table 1: **Results on Wikitext-103** (TNN[59]'s setting). ↓ means *lower is better*.

| Model | PPL (val)↓ | PPL (test)↓ | Params (M) |
|---|---|---|---|
| *Attn-based* | | | |
| Transformer [81] | 24.40 | 24.78 | 44.65 |
| FLASH [10] | 25.92 | 26.70 | 42.17 |
| 1+elu [35] | 27.44 | 28.05 | 44.65 |
| Performer [7] | 62.50 | 63.16 | 44.65 |
| cosFormer [62] | 26.53 | 27.06 | 44.65 |
| *MLP-based* | | | |
| Syn(D) [76] | 31.31 | 32.43 | 46.75 |
| Syn(R) [76] | 33.68 | 34.78 | 44.65 |
| gMLP[42] | 28.08 | 29.13 | 47.83 |
| *RNN-based* | | | |
| S4 [22] | 38.34 | 39.66 | 45.69 |
| DSS [26] | 39.39 | 41.07 | 45.73 |
| GSS [49] | 29.61 | 30.74 | 43.84 |
| RWKV [55] | 24.31 | 25.07 | 46.23 |
| LRU [53] | 29.86 | 31.12 | 46.24 |
| *FFT-based* | | | |
| TNN [59] | 23.98 | 24.67 | 48.68 |
| *Ours* | | | |
| **HGRN** | 24.14 | 24.82 | 46.25 |

# 4 Experiments

We conduct a comparative analysis between our proposed **HGRN** and four widely adopted sequence modeling structures, *i.e.,* attention-based, MLP-based, FFT-based, and state-space-based. We evaluate **HGRN** on the WikiText-103 dataset [50] and the Pile [15] dataset for autoregressive language modeling, as well as the length extrapolation ability. To assess the accuracy and efficiency of our model in handling long-term dependencies, we utilize the LRA benchmark [78]. Additionally, we showcase the robustness of **HGRN** in computer vision task on the ImageNet-1k dataset.

## 4.1 Setting

We implement our models in Pytorch [54] and train them on 8 Nvidia A100 GPUs. For **HGRN**, we found that fusing element-wise recurrence into a single CUDA kernel results in fast running speed in practice. [48] also found that unless the sequence length is sufficiently large, the parallel scan's implementation is not necessarily faster than the sequential scan. As such, we use a CUDA-based sequential scan for implementation; however, our model has the potential to model very long sequences through the use of a parallel scan.

We adopt the same training configuration for all competitors, including batch size, learning rate, training epochs or iterations, *etc.* We list detailed hyper-parameters in the Appendix. For the autoregressive language modeling, we conducted three sets of experiments. Firstly, we validated the performance of two different-scale models on the Wikitext-103 dataset. We used the TNN configuration to verify the performance of the model at around 44m, and the Hyena configuration to verify the performance of the model at around 125m. To evaluate the performance of larger-scale models, we trained a 1b Transformer and **HGRN** on the Pile dataset using 10b tokens. To assess the performance in downstream tasks, we trained **HGRN** models of 150m, 350m, and 1b on the Pile dataset using 100b tokens and conducted zero-shot evaluations on downstream tasks.

For the LRA benchmark, We report results on all 6 tasks. For the image classification on the ImageNet-1k dataset, We integrate **HGRN** into the DeiT [79] structure, we replace the transformer layers with our **HGRN** modules. It is compared to the performance of the vanilla DeiT on the ImageNet-1K dataset for image classification.

## 4.2 Results

Table 4: **Performance Comparison on Commonsense Reasoning.**. PS: parameter size (billion). T: tokens (billion). HS: HellaSwag. WG: WinoGrande.

| Model | Params | Token | BOOLQ | PIQA | HS | WG | ARC-e | ARC-c | OBQA | AVG |
|---|---|---|---|---|---|---|---|---|---|---|
| GPT-Neo | 0.13 | 300 | 61.71 | 63.06 | 30.40 | 50.43 | 43.73 | 23.12 | 26.20 | 42.66 |
| OPT | 0.16 | 300 | 55.47 | 62.95 | 31.35 | 50.43 | 43.52 | 22.70 | 28.00 | 42.06 |
| Pythia | 0.16 | 300 | 55.08 | 61.32 | 30.16 | 51.93 | 43.18 | 23.12 | 26.80 | 41.66 |
| RWKV | 0.17 | - | - | 65.07 | 32.26 | 50.83 | 47.47 | 24.15 | 29.60 | 41.56 |
| HGRN | 0.15 | 100 | 59.91 | 65.02 | 33.33 | 50.20 | 46.68 | 23.81 | 28.60 | 43.94 |
| OPT | 0.35 | 300 | 57.74 | 64.58 | 36.69 | 52.49 | 44.02 | 23.89 | 28.20 | 43.94 |
| Pythia | 0.4 | 300 | 60.40 | 67.08 | 40.52 | 53.59 | 51.81 | 24.15 | 29.40 | 46.71 |
| BLOOM | 0.56 | 350 | 55.14 | 64.09 | 36.97 | 52.80 | 47.35 | 23.98 | 28.20 | 44.08 |
| RWKV | 0.43 | - | - | 67.52 | 40.90 | 51.14 | 52.86 | 25.17 | 32.40 | 45.00 |
| HGRN | 0.35 | 100 | 59.05 | 66.70 | 38.12 | 51.70 | 49.20 | 25.26 | 30.60 | 45.80 |
| GPT-Neo | 1.3 | 300 | 61.99 | 71.11 | 48.93 | 54.93 | 56.19 | 25.85 | 33.60 | 50.37 |
| OPT | 1.3 | 300 | 57.77 | 71.71 | 53.70 | 59.35 | 57.24 | 29.69 | 33.20 | 51.81 |
| Pythia | 1.4 | 300 | 60.73 | 70.67 | 47.18 | 53.51 | 56.99 | 26.88 | 31.40 | 49.62 |
| BLOOM | 1.1 | 350 | 59.08 | 67.14 | 42.98 | 54.93 | 51.47 | 25.68 | 29.40 | 47.24 |
| RWKV | 1.5 | - | - | 72.36 | 52.48 | 54.62 | 60.48 | 29.44 | 34.00 | 50.56 |
| HGRN | 1 | 100 | 58.69 | 70.89 | 48.02 | 51.62 | 55.64 | 27.90 | 31.60 | 49.19 |

**Autoregressive Language Modeling** Autoregressive language modeling stands as a prominent task within the field of natural language processing, as it serves as a measure of a language model's causal inference capability. This task requires the model to estimate the probability distribution of the subsequent token based on the previously seen tokens.

We show the performances of the autoregressive language modeling in table 1 and table 2. Compared to transformer-based methods, **HGRN** performs favourably than most efficient variants of the vanilla transformer such as FLASH [31], 1+elu [35], Performer [7] and cosFormer [62]. Also, **HGRN** achieves better results than the MLP-based methods with a notable margin. Nevertheless, **HGRN** performs similarly to the original transformer [81]. Finally, **HGRN** shares similar concepts with RNN-based such as S4 [22], DSS [26], GSS [49], RWKV [55], and LRU [53], our **HGRN** also achieves superior performance to all RNN-based methods. This provide evidence HRGN may be an effective method in LM We also report the extrapolation ability of **HGRN** compared to previous methods in Table 14.

Table 2: **Results on Wikitext-103** (Hyena[57]'s setting). All models are in GPT-2 small size (125M). ↓ means *lower is better*

| Model | PPL↓ |
|---|---|
| Transformer [57] | 18.6 |
| Hybrid H3 [57] | 18.5 |
| Performer [57] | 26.8 |
| Reformer [57] | 25.6 |
| AFT-conv [57] | 28.2 |
| Linear Attention [57] | 25.6 |
| Hyena [57] | 18.6 |
| Hyena-slim [57] | 18.5 |
| HGRN | 18.6 |

We also trained a 1b model on the Pile dataset and compared it with LRU and Transformer. Specifically, our training parameters included a sequence length of 1024, batch size of 96, 100k updates, and a learning rate of 5e-4. It can be seen that **HGRN** still performs better at the 1b scale. Additionally, we trained 100b tokens of **HGRN** on the Pile dataset at 150m, 350m, and 1b sizes, and evaluated them against open-source Transformer-based models in downstream tasks. We selected Comparison on Commonsense Reasoning and Super GLUE tasks, and all evaluations were done using the lm-evaluation-harness [16]. **HGRN** achieves comparable performance to Transformer-based models when consuming only 1/3 of the tokens.

Table 3: **Results on the Pile.** All the model size is 1b. The lower the better.

| Model | PPL↓ |
|---|---|
| Transformer | 4.56 |
| LRU | 5.07 |
| **HGRN** | 4.14 |

**Long Range Arena** LRA [77] is proposed as a comprehensive evaluation for assessing the performances of models in processing long-term dependencies in various sequential modeling tasks. We show a performance comparison between **HGRN** and existing methods in Table 6. **HGRN** achieves comparable results with other SOTA methods.

Table 5: **Performance Comparison on Super GLUE..** PS: parameter size (billion). T: tokens (billion).

| Model | Params | Token | WSC | WIC | RTE | CB | MULTIRC | BOOLQ | COPA | AVG |
|---|---|---|---|---|---|---|---|---|---|---|
| GPT-Neo | 0.13 | 300 | 36.54 | 50.00 | 54.87 | 41.07 | 0.84 | 61.71 | 64.00 | 44.15 |
| OPT | 0.16 | 300 | 36.54 | 50.00 | 49.82 | 21.43 | 1.36 | 55.47 | 66.00 | 40.09 |
| Pythia | 0.16 | 300 | 36.54 | 50.16 | 52.71 | 41.07 | 2.52 | 55.08 | 65.00 | 43.30 |
| HGRN | 0.15 | 100 | 38.46 | 51.10 | 56.68 | 42.86 | 1.47 | 59.91 | 65.00 | 45.07 |
| OPT | 0.35 | 300 | 36.54 | 50.00 | 51.99 | 46.43 | 1.36 | 57.74 | 72.00 | 45.15 |
| Pythia | 0.4 | 300 | 57.69 | 50.31 | 52.71 | 35.71 | 1.68 | 60.40 | 70.00 | 46.93 |
| BLOOM | 0.56 | 350 | 40.38 | 50.00 | 52.71 | 41.07 | 1.05 | 55.14 | 61.00 | 43.05 |
| HGRN | 0.35 | 100 | 38.46 | 50.16 | 52.71 | 51.79 | 1.99 | 59.05 | 73.00 | 46.74 |
| GPT-Neo | 1.3 | 300 | 36.54 | 50.00 | 60.29 | 44.64 | 1.99 | 61.99 | 69.00 | 46.35 |
| OPT | 1.3 | 300 | 37.50 | 51.10 | 51.99 | 41.07 | 3.15 | 57.77 | 79.00 | 45.94 |
| Pythia | 1.4 | 300 | 36.54 | 50.00 | 53.07 | 35.71 | 0.94 | 60.73 | 72.00 | 44.14 |
| BLOOM | 1.1 | 350 | 36.54 | 50.00 | 52.71 | 41.07 | 0.73 | 59.08 | 68.00 | 44.02 |
| HGRN | 1 | 100 | 40.38 | 50.78 | 53.43 | 42.86 | 3.04 | 58.69 | 70.00 | 45.60 |

Table 6: **Performances Comparison on the Long Range Arena benchmark.** The proposed **HGRN** achieves the best performances and outperforms all competing methods.

| Model | ListOps | Text | Retrieval | Image | Pathfinder | Path-X | AVG. |
|---|---|---|---|---|---|---|---|
| Transformer [81] | 38.37 | 61.95 | 80.69 | 40.57 | 65.26 | - | 47.81 |
| cosFormer [62] | 36.50 | 67.70 | 83.15 | 51.23 | 71.96 | - | 51.76 |
| FLASH [31] | 38.70 | 64.10 | 86.10 | 47.40 | 70.25 | - | 51.09 |
| S4 [22] | 59.60 | 86.82 | 90.90 | 88.65 | 94.20 | 96.35 | 86.09 |
| DSS_softmax [26] | 60.60 | 84.80 | 87.80 | 85.70 | 84.60 | 87.80 | 81.88 |
| DSSEXP [26] | 59.70 | 84.60 | 87.60 | 84.90 | 84.70 | 85.60 | 81.18 |
| DSSEXP-NO-SCALE [26] | 59.30 | 82.40 | 86.00 | 81.20 | 81.30 | - | 65.03 |
| TNN [59] | 61.04 | 87.90 | 90.97 | 88.24 | 93.00 | 96.10 | 86.21 |
| S5 [71] | 62.15 | 89.31 | 91.4 | 88 | 95.33 | 98.56 | 87.46 |
| Mega [46] | 63.14 | 90.43 | 91.25 | 90.44 | 96.01 | 97.98 | 88.21 |
| SGConv [41] | 61.45 | 89.2 | 91.11 | 87.97 | 95.46 | 97.83 | 87.17 |
| LRU [53] | 60.20 | 89.40 | 89.90 | 89.00 | 95.10 | 94.20 | 86.30 |
| **HGRN** | 59.95 | 88.14 | 94.23 | 88.69 | 92.92 | 97.50 | 86.91 |

**Image Classification**    The image classification results on the ImageNet-1K dataset are presented in Table 7. Notably, with comparable parameter sizes, our proposed **HGRN** model demonstrates superior performance compared to previous methods such as TNN and the vanilla transformer. It demonstrates the capability of **HGRN** in modeling visual modalities.

Table 7: **Performances comparison of image classification on ImageNet-1k. HGRN** performs favorably than competing methods with similar parameter sizes.

| | DeiT-Tiny | | DeiT-Small | |
|---|---|---|---|---|
| Model | Top1 Acc | Param (M) | Top1 Acc | Parma (M) |
| Deit | 72.20 | 5.7 | 79.90 | 22.0 |
| TNN | 72.29 | 6.4 | 79.20 | 23.4 |
| **HGRN** | 74.40 | 6.1 | 80.09 | 23.7 |

## 4.3   Ablation Study

We conducted ablation studies in the smallest-scaled setting (i.e., TNN[59]'s setting on Wiki-Text103 dataset) to thoroughly verify the effectiveness of each of our proposed components in **HGRN**. Experiments were conducted on the Pile dataset using a 1b model with 10b tokens for the forget gate experiment.

Table 8: **Forget gate ablation** on an autoregressive language model. The only lower bound means using a data-independent gate like LRU.

| Model | PPL↓ |
|---|---|
| LRU w forget gate | 4.92 |
| LRU | 5.07 |
| **HGRN** only lower bound | 4.84 |
| **HGRN** w/o forget gate | 57.42 |
| **HGRN** | 4.14 |

Figure 2: **Visualization of forget rates.** We plot the forget rates of layers 5 and 6 on a model trained on language modeling tasks.

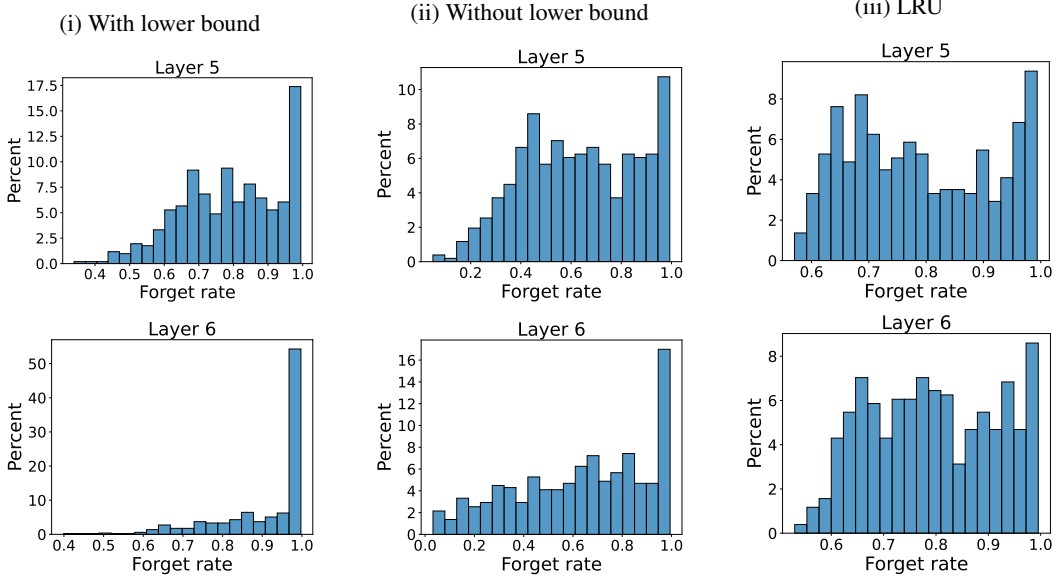

**The influence of forget gate**   In table 8, we demonstrate the role of forget gate. From table 8, we observe that removing the forget gate significantly decreases the performance of **HGRN**, while adding a forget gate to LRU improves performance. On the other hand, using a data-independent forget gate (only lower bound) leads to lower performance compared to a data-dependent forget gate.

**The influence of input gate and output gate**
Table. 9 validates the effectiveness of using output gates and tying input and forget gates. w/o input gate means to remove the $1 - \lambda_t$ term. w/o output gate means remove the left branch of **HGRN** in figure 1. Our design achieves the best performance.

Table 9: **Ablations of gates on autoregressive language modeling.** w/o input gate means to remove the $1 - \lambda_t$ term. w/o out_gate means remove the left branch of **HGRU** in figure 1.

| Model | PPL↓ |
|---|---|
| w/o input gate | 25.03 |
| w/o output gate | 25.50 |
| **HGRN** | 24.14 |

**The influence of lower bounds in forget gate values**   We demonstrate the effectiveness of introducing a lower bound in Table 10 and Table 13. From Table 10, we observe that gating (i.e., without lower bound) is more critical than the lower bound (i.e., only lower bound). Combining gating and the lower bound consistently provides benefits, but the most significant improvement arises from the monotonically increasing lower bound. This aligns with the intuition that lower layers should primarily focus on nearby tokens, while upper layers should attend more broadly to capture long-term dependencies [58].

Table 13 highlights the essential role of the lower bound in long sequence processing tasks. In these tasks, the model's performance is notably poor and sometimes fails to converge without the lower bound. It is worth noting that language modeling tasks do not require extensive long-term dependencies, which explains why the model performs well even without the lower bound. However, in the task of LRA, the ability to capture long-term dependencies is crucial for achieving satisfactory performance.

Table 10: **Lower bound ablation** on autoregressive language modeling. A random lower bound means the lower bound in each layer is independent. Decrease lower bound means the lower bound is monotonically decreasing with respect to layer $k$, only the lower bound means the forget rate is independent of input.

| Model | PPL ↓ |
|---|---|
| w/o lower bound | 24.71 |
| random lower bound | 24.60 |
| decrease lower bound | 24.63 |
| only lower bound | 27.70 |
| **HGRN** | 24.14 |

**The influence of complex-valued recurrence**
Table 11 validates the utility of incorporating

complex values in element-wise linear recurrence. Additionally, the experiments show that the phase argument $\theta$ should not be data-dependent.

### 4.4 Analysis on forget gate values

We present the distributions of forget gate values across layers for different methods in Table 12 and visualize the histogram of each layer in Figure 2, trained on the autoregressive language modeling task. The results demonstrate that the addition of lower bounds effectively increases the average forget gate values in higher layers (5-6). Notably, the medium forget gate values in the highest layer reach 0.98, enabling the modeling of long-term dependencies.

Table 11: **Ablations of complex-valued recurrence on autoregressive language modeling.** w/o complex means remove theta, data-dependent theta means theta is dependent on the input, this makes the matrix $\Lambda$ not a Toeplitz matrix, which can not capture relative information.

| Model | PPL$\downarrow$ |
|---|---|
| w/o complex | 25.34 |
| data dependent $\theta$ | 28.74 |
| **HGRN** | 24.14 |

It is interesting to note that the average forget gate values of the LRU model consistently exceed those of our variant model without lower bounds, as per their eigenvalues. However, despite this, the language modeling performance of LRU is lower than that of our variant. Specifically, LRU scored 24.71, while our variant scored 31.12. This suggests that using data-dependent gates to selectively retain relevant information is advantageous, rather than relying on data-independent forget gate values across all time steps.

Table 12: **Forget gate values of different methods on language modeling tasks.** In each layer, we counted the mean and median of forget gate values.

| Layer | ours mean | ours median | w/o lower bound mean | w/o lower bound median | LRU mean | LRU median |
|---|---|---|---|---|---|---|
| 1 | 0.48 | 0.47 | 0.52 | 0.50 | 0.75 | 0.72 |
| 2 | 0.55 | 0.52 | 0.59 | 0.55 | 0.78 | 0.75 |
| 3 | 0.60 | 0.57 | 0.58 | 0.56 | 0.78 | 0.76 |
| 4 | 0.68 | 0.64 | 0.58 | 0.55 | 0.79 | 0.78 |
| 5 | 0.79 | 0.80 | 0.63 | 0.63 | 0.79 | 0.77 |
| 6 | 0.91 | 0.98 | 0.63 | 0.67 | 0.79 | 0.79 |

Table 13: **Lower bound ablation on LRA.** We verify the importance of lower bounds in long-sequence modeling capabilities.

| Model | ListOps | Text | Retrieval | Image | Pathfinder | Path-X | AVG |
|---|---|---|---|---|---|---|---|
| w/o lower bound | 51.41 | 87.79 | 88.71 | 80.17 | - | - | 51.53 |
| **HGRN** | 59.95 | 88.14 | 94.23 | 88.69 | 92.92 | 97.50 | 86.91 |

## 5 Conclusion

In this work, we have shown that gated linear RNNs could obtain impressive performance across different tasks and modalities without compromising efficiency. We highlighted the significance of the forget gate for linear RNNs in language modeling and emphasized the importance of an additive lower bound on forget gate values for modeling long-term dependencies.

## Acknowledgement

This work is partially supported by the National Key R&D Program of China (NO.2022ZD0160100).

## Limitations and broader impact

Our empirical evaluation of **HGRN** remains on a smaller scale compared to other large-scale models. Potentially negative social consequences include the misuse of brain models for unsuitable purposes or applications, which must be prohibited by appropriate rules. In the era of large language models, the inference cost is the key limitation of transformer-based models. RNNs provide a solution with their lower inference costs. This could potentially lead to a significant evolution in the field.

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

# 6 Appendix

In this appendix, we examine the extrapolation ability of **HGRN** and provide the training and inference speed comparison of **HGRN** and existing efficient sequence modeling methods. We also illustrate the forget rates of each layer on a trained language model of **HGRN**.

We also report the extrapolation ability of **HGRN** compared to previous methods in Table 14.

## 6.1 Extrapolation test

In this section, we tested **HGRN** 's extrapolation ability by directly inferring the model with a variety of sequence lengths. As shown in Table 14, our method has the ability to train short and test long.

## 6.2 Speed comparison

In this section, we benchmark the speed of our method on the LRA benchmark. Our method achieves state-of-the-art training and inference speed.

## 6.3 Visualization

In this section, we visualize the forget rates of each layer on a model trained on language modeling tasks.

## 6.4 Configurations

We list detailed hyper-parameters of our experiments here.

Figure 3: Visualization forget rates in each layer.

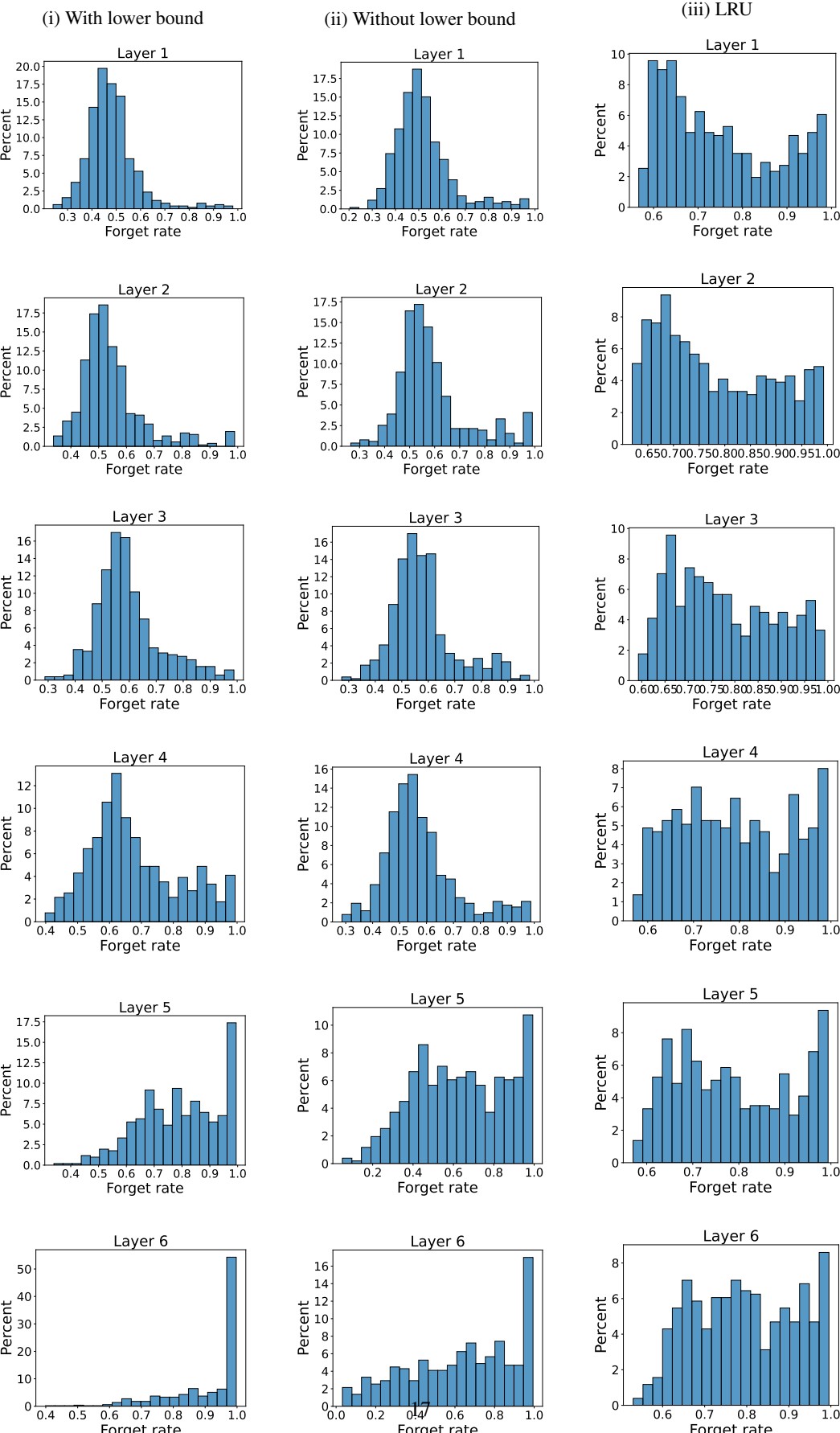

Figure 4: Visualization of token mixing matrix in each layer.

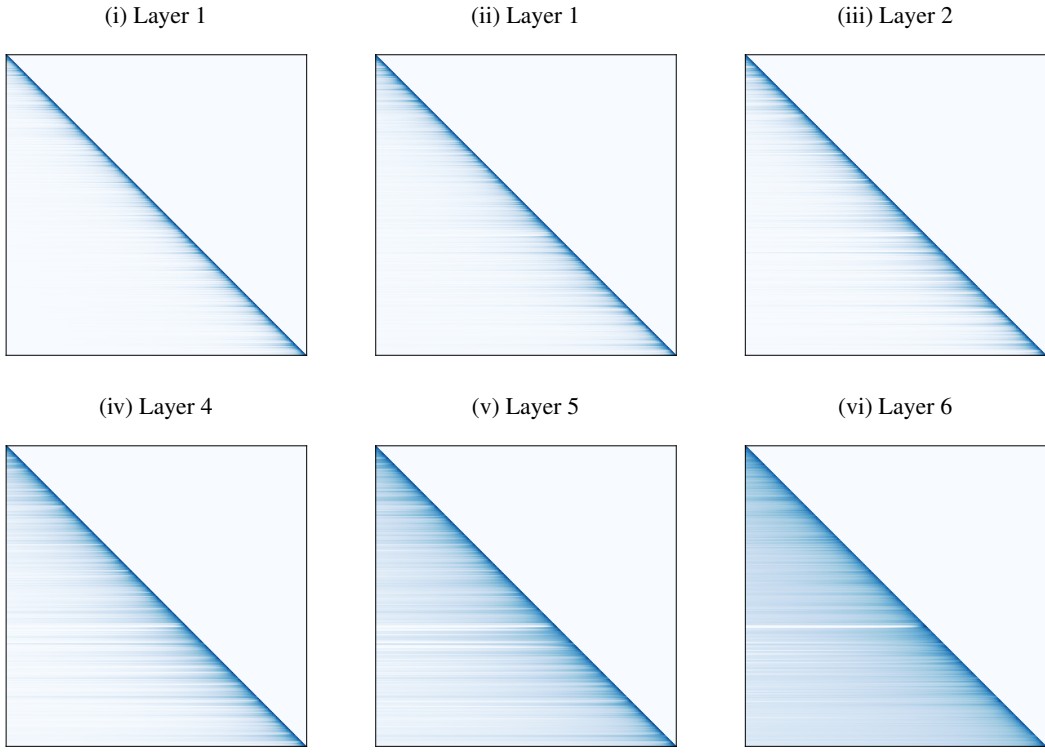

(i) Layer 1     (ii) Layer 1     (iii) Layer 2

(iv) Layer 4     (v) Layer 5     (vi) Layer 6

Table 14: The extrapolation performance of competing methods. The best result is highlighted in **bold** and the second in underline. ↓ means *lower is better*.

| Seqlen | Transformer PPL↓ | LS PPL↓ | FLASH PPL↓ | 1+elu PPL↓ | Performer PPL↓ | cosFormer PPL↓ | gMLP PPL↓ | S4 PPL↓ | DSS PPL↓ | GSS PPL↓ | ALiBi PPL↓ | TNN PPL↓ | LRU PPL↓ | **HGRU** PPL↓ |
|---|---|---|---|---|---|---|---|---|---|---|---|---|---|---|
| 512 | 24.78 | 24.05 | 24.69 | 28.05 | 63.16 | 27.06 | 29.13 | 30.74 | 41.07 | 39.66 | 24.15 | 24.67 | 31.12 | 24.85 |
| 768 | 41.36 | 23.49 | 16950.45 | 47.35 | 159.74 | 32.90 | 1.34E+9 | 30.41 | 40.50 | 39.76 | 23.38 | 24.25 | 30.72 | 24.4 |
| 1024 | 62.35 | 23.21 | 174165.47 | 70.47 | 504.30 | 55.28 | 8.93E+12 | 30.24 | 40.22 | 39.91 | 22.98 | 24.05 | 30.5 | 24.16 |
| 1280 | 82.52 | 23.07 | 346502.88 | 91.88 | 1020.28 | 102.88 | 1.58E+15 | 30.15 | 40.03 | 40.82 | 22.74 | 23.91 | 30.38 | 24.03 |
| 1536 | 100.17 | 22.97 | 647788.12 | 111.56 | 1568.83 | 175.26 | 4.96E+16 | 30.08 | 39.94 | 41.04 | 22.57 | 23.83 | 30.3 | 23.94 |
| 1792 | 118.42 | 22.97 | 1719873.5 | 129.92 | 2138.50 | 267.65 | 5.67E+17 | 30.04 | 39.85 | 41.08 | 22.52 | 23.79 | 30.24 | 23.88 |
| 2048 | 133.44 | 22.99 | 6.25E+6 | 147.09 | 2693.89 | 368.02 | 3.59E+18 | 30.00 | 39.79 | 41.53 | 22.43 | 23.73 | 30.19 | 23.82 |
| 3072 | 188.95 | 23.25 | 4.17E+10 | 206.88 | 4945.82 | 820.77 | 2.19E+20 | 29.91 | 39.64 | 44.08 | 22.24 | 23.63 | 30.09 | 23.71 |
| 4096 | 246.06 | 23.83 | 2.67E+13 | 267.87 | 7170.91 | 1335.51 | 1.61E+21 | 29.88 | 39.59 | 48.27 | 22.17 | 23.58 | 30.04 | 23.66 |
| 5120 | 270.93 | 24.56 | 1.26E+15 | 299.31 | 8443.15 | 1735.50 | 5.08E+21 | 29.85 | 39.54 | 53.32 | 22.11 | 23.54 | 30.01 | 23.62 |
| 6144 | 311.65 | 25.45 | 1.58E+16 | 352.62 | 10234.07 | 2146.19 | 1.16E+22 | 29.83 | 39.51 | 57.73 | 22.08 | 23.53 | 29.99 | 23.6 |
| 7168 | 346.58 | 26.42 | 8.11E+16 | 389.02 | 11420.56 | 2494.79 | 1.98E+22 | 29.82 | 39.49 | 60.25 | 22.07 | 23.51 | 29.97 | 23.58 |
| 8192 | 372.18 | 27.11 | 3.40E+17 | 411.50 | 12557.09 | 2902.24 | 2.78E+22 | 29.82 | 39.49 | 63.36 | 22.05 | 23.51 | 29.97 | 23.58 |
| 9216 | 387.29 | 28.78 | 1.22E+18 | 453.27 | 14847.66 | 3028.72 | 3.93E+22 | 29.80 | 39.46 | 74.92 | 22.03 | 23.49 | 29.96 | 23.56 |
| 10240 | 395.94 | 30.13 | 4.03E+18 | 457.06 | 13623.83 | 3247.83 | 4.93E+22 | 29.79 | 39.45 | 81.87 | 22.02 | 23.48 | 29.94 | 23.55 |
| 11264 | 426.54 | 31.14 | 1.07E+19 | 504.19 | 14661.77 | 3341.91 | 5.70E+22 | 29.79 | 39.46 | 87.67 | 22.00 | 23.48 | 29.94 | 23.55 |
| 12288 | 463.50 | 33.21 | 2.52E+19 | 555.38 | 17959.85 | 3644.81 | 7.18E+22 | 29.79 | 39.44 | 92.11 | 22.00 | 23.48 | 29.94 | 23.55 |
| 13312 | 506.35 | 34.72 | 4.96E+19 | 584.01 | 20026.35 | 3851.70 | 8.04E+22 | 29.78 | 39.43 | 96.00 | 22.00 | 23.47 | 29.93 | 23.54 |
| 14336 | 486.86 | 36.05 | 1.28E+20 | 589.83 | 20971.31 | 3951.26 | 9.41E+22 | 29.78 | 39.43 | 101.47 | 21.99 | 23.46 | 29.92 | 23.53 |
| Avg | 261.36 | 26.71 | 1.16E+19 | 299.86 | 8684.79 | 1764.75 | 2.41E+22 | 29.97 | 39.75 | 60.26 | **22.40** | 23.70 | 30.17 | 23.80 |

Table 15: Speed comparison on LRA benchmark. The 1K,...,5K represent the input sequence length. We mark it with - if a method is out of memory. The higher the better for all metrics.

| Method | Train Speed(steps per second)↑ | | | | | Inference Speed(steps per second)↑ | | | | |
|---|---|---|---|---|---|---|---|---|---|---|
| | 1K | 2K | 3K | 4K | 5K | 1K | 2K | 3K | 4K | 5K |
| Transformer [81] | 13.58 | 4.84 | - | - | - | 23.67 | 8.22 | - | - | - |
| Performer [36] | 18.40 | 10.77 | 7.66 | 6.30 | 5.64 | 30.04 | 17.36 | 12.80 | 10.55 | 9.52 |
| LS [85] | 20.29 | 11.24 | 8.05 | 6.51 | 5.89 | 39.05 | 21.11 | 15.02 | 12.6 | 11.66 |
| Fnet [39] | 25.19 | 15.62 | 11.24 | 9.41 | 8.18 | 48.81 | 27.89 | 19.52 | 16.27 | 14.46 |
| cosFormer [62] | 22.00 | 12.80 | 9.47 | 7.93 | 7.13 | 39.05 | 22.31 | 16.62 | 13.95 | 12.60 |
| S4 [21] | 13.13 | 7.33 | 4.91 | 3.84 | 3.04 | 30.04 | 16.27 | 10.85 | 8.58 | 6.79 |
| FLASH [31] | 17.36 | 9.03 | 6.54 | 5.19 | 4.68 | 30.04 | 15.94 | 11.32 | 9.19 | 8.40 |
| TNN [59] | 17.55 | 9.89 | 6.79 | 5.68 | 4.54 | 33.96 | 17.75 | 12.40 | 10.28 | 8.22 |
| **HGRU** | 22.31 | 13.58 | 9.52 | 7.40 | 7.44 | 43.39 | 25.19 | 16.62 | 14.20 | 13.95 |

Table 16: Detailed training configurations used in our experiments. "Total batch size" means batch_per_gpu × update_freq × num_gpus. "ALM" stands for Autoregressive Language Model. "IM" stands for Image Modeling.

| | AML | IM |
|---|---|---|
| Data | WikiText-103 | ImageNet-1k |
| Tokenizer method | BPE | - |
| Src Vocab size | 50265 | - |
| Sequence length | 512 | - |
| Total batch size | 128 | 2048 |
| Number of updates/epochs | 50k updates | 300 epochs |
| Warmup steps/epochs | 4k steps | 5 epochs |
| Peak learning rate | 5e-4 | 2.5e-4 |
| Learning rate scheduler | Inverse sqrt | cosine |
| Optimizer | Adam | Adamw |
| Adam $\epsilon$ | 1e-8 | 1e-8 |
| Adam $(\beta_1, \beta_2)$ | (0.9, 0.98) | (0.9, 0.98) |
| Weight decay | 0.2 | 0.1 |
| Gradient clipping | - | 1.0 |

