# OpenReview forum: "Hierarchically Gated Recurrent Neural Network for Sequence Modeling"
_NeurIPS.cc/2023/Conference — NeurIPS 2023 spotlight_

### Official Review · Reviewer_QJYn · 2023-06-10

**Soundness:** 2 fair
**Presentation:** 1 poor
**Contribution:** 3 good
**Rating:** 6
**Confidence:** 5

**Summary:**

This work is concerned with improving linear RNNs (which can be efficiently parallelized) for long-sequence modeling. It proposes to address this problem with gating schemes and complex forget gate values. They introduce a learnable lower bound on the forget gate (or essentially the diagonal dynamics matrix) for each layer to capture both short-term and long-range dependencies. The result is the proposed Hierarchical Gated Recurrent Units (HGRU). This RNN is evaluated on Wikitext, LRA and Imagenet-1K. Ablations are performed to analyze the effectiveness of the design choices.

**Strengths:**

- The area of linear SSMs/RNNs has gained a fair amount of recent attention and the angle this paper takes of considering the linear RNN dynamics matrix from the point of view of the LSTM forget gate is of interest to the community and has the potential to help bridge the gap between prior RNN work and the more modern linear SSM/RNN work
- While not framed in this way, the forget gate formulation serves as a practical example of input-dependent dynamics, of interest to the deep SSM community
- The method appears to achieve strong performance on the tasks evaluated in this paper
- The ablations are helpful to understand the design choices better

**Weaknesses:**

- While most relevant works are cited in some place in the paper, the results of many of these methods are inexplicably left out of the results tables, leading to misleading statements:
  - For example SgConv and MEGA are both cited in the paper, but their results are left out of the LRA table. Both of these methods have a better Path-X score (97.83 and 97.98) and overall average (87.17 and 88.21) than the proposed HGRU and both have been released on Arxiv since 2022. This makes the statements in lines 229 that HRGU "is better than all previous methods" and lines 231-232 that HRGU "outperforms all previous methods on this task"  clearly incorrect.
  - Similarly, S5 (https://arxiv.org/abs/2208.04933) is never cited at all in the paper despite the fact that it is a linear RNN method that uses the same parallel scan approach for computing a linear recurrence as HGRU (assuming HGRU actually uses a parallel scan though see questions for clarification). The LRU paper that is cited discusses S5 extensively.   In addition, S5's scores are left out of the LRA table despite having a better path-X and overall average than HGRU (98.58 and 87.46), similar to the SgConv and Mega examples mentioned above.  This is particularly relevant since it serves as an ablation showing the forget gate innovations in this paper are not strictly necessary for strong longe-range modeling with a linear RNN.
  - To clarify, I do not think it would hurt the story of this paper to include these methods that perform slightly better, and in fact I think the statement in line 232 "It validates the ability of HRGN in handling long sequence tasks" is completely reasonable and supported by the evidence. It is just the exclusion of these other methods may mislead readers not as familiar with the research on what is strictly necessary for strong performance and the statements listed above regarding "outperforms all..." are incorrect.
  - Finally, given LRU is cited as concurrent work, it would also be relevant to cite DLR (https://arxiv.org/abs/2212.00768) since it explores similar themes.

- It would seem that H3 (cited in the paper) should have been a baseline for the Wikitext-103 experiments. Alternatively, the LRU could have been plugged in for the S4D SSMs used in H3.  As stated in the paper, it is well known in the literature at this point that just using an SSM/Linear RNN such as S4, DSS or LRU without additional gating is not sufficient for language modeling. But additional gating such as in H3 has been shown to work well, such as in the H3 paper and the Wang et al. 2022 papers that are cited. Including this baseline would allow for a better comparison with prior methods on language and allow for a better understanding of the effectiveness of the HGRU innovations vs the gating findings from previous work.

- The overall presentation is poor. Many of the tables say that certain scores are bold or underlined, yet the tables often have no bolds or underlines, or it is inconsistent. Many things are promised to be in the appendix, but then are not there (see questions section below for specific examples). I think some of the sections in the presentation of the method could be improved (see questions section below for suggestions). In addition, Figure 1 is not clearly labeled, leaving one guessing on which gates are located where.

- The authors have checked yes on the author checklist for including training details and reproducibility, but I think this is clearly wrong. First, it is not even clear which methods were implemented by the authors and which methods have numbers reported from other works. Section 4.1 states that "we adopt the same training configuration for all competitors...." and that detailed hyperparams are listed in the appendix. However, no hyperparameters are listed in the appendix and many of the performance scores appear to be reported directly from  the prior papers. In addition, no code is included to allow one to try and make sense of this. The authors should make it clear which methods were implemented and evaluated as well as provide detailed hyperparameters to enable reproducibility.

**Questions:**

Major questions/suggestions:

- Line 186: It is stated here that $\lambda_t$'s near 1 "makes HRGN more efficient modeling of global information". What does efficient mean here? More efficient than what? How is this claim supported? More clarification should be provided here to avoid making an imprecise or unwarranted claim.

- Line 187: The "visualization of the token mixing matrix" cannot be found in the appendix

- Section 3.4: I would recommend moving this section up closer to the "Lower bound on forget gate values" section. I found myself having to scroll back and forth to remind myself of the definitions.

- Figure 1: needs more labels. How many gates are there? Is the output gate from equation 3 located in the HRU (per the comment in line 174) or in the HGRU (middle figure)? Clearly labeling the location of the f and g gates, as well as the hidden state h would help a ton.  It would also be nice if the full equation for the recurrence (including the computation of $\lambda_t$) was a part of this diagram, where one would not have to wonder what each of the different symbols mean, or scroll throughout the paper to piece this together.

- Algorithm 1: I would recommend including a line for the explicit computation of $\lambda_t$. This is an important part of your approach, but the algorithm makes it appear like this is just some learnable parameter.

- Line 208: What does identical training hyperparameters mean? It is not even clear the baselines reported from other papers all use the same hyperparameters. Also is this the right approach? It is not obvious that each method should use the exact same learning rate etc and I would think it would be reasonable to perform a mild sweep over a few different learning rates and other settings. If not, how did you choose the learning rate? Did you choose the learning rate that was best for HRGN? More information here would allow a reader to better assess the reported results.

- Line 224: "proves the effectiveness of our method in LM." I frankly think this is too strong of a claim. It is well known that performance on wikitext-103 can be sensitive to regularization since the dataset is relatively small (at least in 2023). I think a more accurate and defensible statement might read as these results "provide evidence HRGN may be an effective method in LM". I also personally think this weakened claim is sufficient for an interesting paper.

- Line 274: I think the discussion of the "forget gate values" of LRU is interesting but could be further developed and discussed. This provides an opportunity to connect the forget gates discussed in this paper with the dynamics state matrices used by S4/S5/LRU etc. In addition, I think the point in line 280 about data-dependent gates (or dynamics matrices) is interesting. It would be relevant to cite Liquid-S4   (https://arxiv.org/abs/2209.12951) here since they propose a method to have input-dependent dynamics with an S4-like approach. This approach also has strong performance on LRA and their results should also be included.

- How is the recurrence actually computed? Are parallel scans used? A parallel scan is mentioned in line 135 when talking about the generic equations before the HGRU is fully introduced. Is this how this is computed? Was the parallel scan implemented in Torch? Also, in line 136, Martin and Cundy should probably be cited here for the parallel scan since they were one of the first to suggest their use for linear RNNs. S5 could also be cited since it was the first to actually make this approach work well for challenging long-range problems. LRU could also be cited since they show an effective method that uses a parallel scan.

- More details should be provided for the speed comparisons in Table 10. Also, how does it scale to longer sequences than 5K, e.g. the 16K of Path-X? How does this compare to other methods that use a parallel scan such as S5 or LRU? Or other diagonal RNNS/SSMs that use convolutions such as H3/S4D/DSS? S4 times are listed here, but I wonder if the optimized Cauchy kernel implementation was used? It is also unclear to me why HGRU would be significantly faster than S4 at inference speed? Both are run as RNNs? Is this due to hyperparameter/state size selection? How was this chosen? More details and comparisons would resolve these questions.

Other questions/suggestions:
- Line 39: the LSSL (https://arxiv.org/abs/2110.13985) and S4 papers should be cited here since these were the first papers that showed this arrangement of linear layers and nonlinear activations can work well on challenging tasks at scale.

- Line 68: S4D should be cited here since it was the first work to show a significant performance increase  for linear SSM/RNN methods over the original S4 release by using GLU activations.

- Line 72: It might make since to cite S4, S4D, DSS, Liquid S4 (https://arxiv.org/abs/2209.12951) here also since they are all long convolution methods

- Line 85: While [3] and [37] both suggested removing the nonlinearity on the recurrence, it really was not until the LSSL and S4 papers that it was shown you can actually do this without losing performance. These methods should probably be cited here.

- Line 91: S5 also uses a diagonal recurrence matrix and could be cited here

- Line 102: MEGA very explicitly uses an EMA approach and could be cited here

- Line 105: in regards to LRU "we illustrate that the addition of gating scheme substantially enhances its performance", but this is basically what is already known from H3. So a good baseline for the Wikitext table would have been to either compare to H3 using S4D, or to compare to LRU plugged into H3. This would allow for a comparison with the other differences between this H3-LRU variant and the HGRU.

- Line 142:  Along with the other cited approaches. DLR and S5 should also be cited here as linear RNNs that uses complex-valued eigenvalues.

- Line 143: I think $g_t$ is introduced here without explanation. Presumably this refers to the output gate described around line 172, but is confusing when it has not been introduced yet

- Line 151: I would not use the phrase "final formulation" here, when there is much more formulation to be discussed. This confused me at first because I was trying to figure out how $\lambda_t$ was determined and had to read on to realize it was not a data-independent learnable parameter.

- Line 156: ON-LSTM is mentioned here without much explanation. Why is this relevant? A bit more explanation here might be helpful.

- Line 174: Has $\tau$ been introduced? What kind of nonlinearity is this?

- Line 292: typo, should be broader impact and not "border" impact


In summary, I like the main ideas of this paper, think it has the potential to be a strong submission and would like to see it accepted. But I do not think it can be accepted in its current form. However, I do think it is feasible to incorporate the suggestions above and I am willing to revise my score upward.

**Limitations:**

Limitations are discussed.

---

> ### Author Rebuttal · Authors · 2023-08-09
>
> **Q1: More citations and experimental results**.
>
> A1: We will include the results for SgConv, MEGA, and S5 in the revised version and rectify our narrative accordingly, and add citation for DLR. We will also include the results for S5 in the upcoming update. Regarding the forget gate, we conducted additional experiments using the LM configuration from Hyena. The previous experimental results for HGRN are from [1]. From the table below, it is evident that our method performs comparably to Transformer, Hyena, and H3 in terms of performance. On the other hand, the performance drop is substantial when the forget gate is removed. This validation underscores the significant role of the forget gate.
>
> | Model | PPL |
> | --- | --- |
> | Transformer | 18.6 |
> | Hybrid H3 | 18.5 |
> | Performer | 26.8 |
> | Reformer | 25.6 |
> | AFT-conv | 28.2 |
> | Linear Attention | 25.6 |
> | Hyena-Convolution | 18.6 |
> | Hyena-slim-Convolution | 18.5 |
> | HGRN | 18.6 |
> | HGRN w/o forget gate | 20.0 |
> | HGRN w/o lower bound | 19.1 |
>
> Besides, the intention of our paper is not to assert an **absolute necessity** for forgetting gates in linear RNNs for long sequence modeling. Rather, our primary objective is to illustrate that simple gated RNNs could exhibit remarkable performance in LRA through the introduction of a lower bound ( in Table 8). **We remark that HGRU is conceptually much simpler than prior state-space models. It obviates the need for meticulously crafted initializations (e.g., Hippos) and yet demonstrates superior performance in real-world tasks such as language modeling.**
>
> **Q2: More experimental results on language modeling.**
>
> A2: We utilize the codebase from H3 to conduct experiments on a GPT-small scale model (～125M parameters) for our HGRN. THe result is listed in R4Q1. Remarkably, HGRN achieves performance parity with Transformers **without incorporating any attention layers**, unlike H3 which employs two such layers. We also scaled our model to 1B parameters and trained on The Pile dataset. As shown in R3 Q1,  HGRN outperforms the Transformer by a clear margin.
>
> **Q3: The presentation quality.**
>
> A3: We will fix these issues in the updated versions.
>
> **Q4: Checklist for including training details and reproducibility**
>
> A4: We will include in the subsequent update a breakdown of which methods were implemented by us and which were referenced. Our configurations are from the Toeplitz neural network [2] and are listed in `one-page PDF`. We will release the code to ensure reproducibility.
>
> **Q5: What meaning of efficient?**
>
> A5: This is a typo. It should be corrected to 'effective'.
>
> **Q6: Visualization of the token mixing matrix.**
>
> A6: We have already added the results of the visualization, as demonstrated in `one-page PDF`.
>
> **Q7: The location of "Lower bound on forget gate values" section, more labels on Figure 1 , more detail on Algorithm 1.**
>
> A7: We will fix this issue in the revised versions.
>
> **Q8: Training configurations.**
>
> A8: We retained the configuration of the Toeplitz neural network as described in reference [1], without performing any hyperparameter adjustments.
>
> **Q9: Too strong of a claim.**
>
> A9: Thanks for the suggestion; we will modify this statement in the updated version accordingly.
>
> **Q10: Citation of Liquid-S4.**
>
> A10: We will add the results for Liquid-S4 in the updated versions.
>
> **Q11: Implementation of recurrence.**
>
> A11: We implement linear recurrence by fusing the entire for loop into a single CUDA kernel, akin to RWKV's implementation. Reference to Table 1 in [3] indicates that parallel scan is only faster when sentence lengths exceed 10k, so we did not use parallel scan in training. Still, our model admits the use of the parallel scan, which could be potentially useful in domains such as RNA modeling.
>
> **Q12: More details of the speed comparisons in Table 10.**
>
> A12: In Table 10, we compare various methods based on their speed results with similar parameter counts. Our approach uses a loop-based CUDA implementation, and for the version with a length of 16K, the parallel scan could be faster. The speed of LRU is expected to be similar to ours. Our training speed is notably faster than S4 because of the I/O bottleneck caused by Torch's FFT. We used the official S4 implementation that employs the Cauchy kernel. The speed of S4 is determined by its official code implementation.
>
> **Q13: More citations such as LSSL, S4D, etc.**
>
> A13: We will add the references in the reversion versions.
>
> **Q14: Comparation to H3.**
>
> A14: Please Refer to R4 A1.
>
> **Q15: $g_t$ is introduced here without explanation**.
>
> A15: We will address this issue in the reversion versions.
>
> **Q16: Do not use the phrase "final formulation".**
>
> A16: We will address this issue in the reversion versions.
>
> **Q17: Question aboud ON-LSTM.**
>
> A17: The relevance lies in its introduction of the `cummax` function. However, our utilization of the `cummax` function differs, as explicated in line 158: our objective is to drive the forget gate values of higher layers towards one, in contrast to the desire for multiscale forget gate values within a single layer in ONLSTM.
>
> **Q18: Typos.**
>
> A18: We will correct the typo in the revised version.
>
> Citations:
> [1] Albert Gu, Caglar Gulcehre, Thomas Paine, Matt Hoffman, Razvan Pascanu; Proceedings of the 37th International Conference on Machine Learning, PMLR 119:3800-3809
> [2]  Zhen Qin, Xiaodong Han, Weixuan Sun, Bowen He, Dong Li, Dongxu Li, Yuchao Dai, Lingpeng Kong, and Yiran Zhong. Toeplitz neural network for sequence modeling. In The Eleventh International Conference on Learning Representations, 2023.
> [3] Eric Martin and Chris Cundy; Parallelizing Linear Recurrent Neural Nets Over Sequence Length; In The Six International Conference on Learning Representations, 2018.

---

> > ### Comment · Reviewer_QJYn · 2023-08-11
> >
> > Thank you for your rebuttal. I do hope all of the citation and presentation issues raised by myself and other reviewers will be revised as promised.
> >
> > **Re. "The previous experimental results for HGRN are from [1]"**: Perhaps this was a typo, but what experiments is this in reference to? I see very little overlap between the experiments in the cited paper and the experiments in the HGRN paper. My broader point was that it is unclear which baselines have numbers reported from other works, and which baselines were rerun by the authors. As well as how hyperparameters compare, were selected, etc. This should all be made more clear.
> >
> > **Wikitext Ablation:** I appreciate the authors including this new setup for the Wikitext experiments. But I am not sure I find the forget gate ablation here incredibly convincing. Wikitext-103 is relatively small and easy to overfit, even with the 125M model size. So small changes in perplexity can be simply due to different hyperparameters/regularization needs.
> >
> > **Re. "intention of our paper is not to assert an absolute necessity for forgetting gates in linear RNNs...our primary objective is to illustrate that simple gated RNNs could exhibit remarkable performance in LRA through the introduction of a lower bound ":** I agree this is an interesting scientific question to investigate. But the paper would be much higher impact if it illustrated the necessity or value of the forget gates, at least for some tasks.  The existence of RNN/SSM methods that can perform well or better on LRA and language modeling without forget gates limits the impact of this paper.
> >
> > **Simplicity:** I would be careful with how much you try to attach the paper's value to the method's simplicity, since this can be quite subjective. E.g. one might argue that having to carefully think about and choose lower bound values for each layer is not simple. On the other hand, trying to contrast the simplicity of this approach to "Hippos" is a bit of a straw man, since the diagonal SSM methods (e.g. DSS, S4D, S5, LRU) all make the formulation much simpler than the original cauchy kernel computations required in the original S4 work. I am not making a judgement about how simple or complicated HGRN is, merely pointing out how subjective this can be.
> >
> > **New wikitext results:** It is good that HGRN achieves parity with Transformers for 125M models on Wikitext-103. However, it is not so remarkable that it did this without attention layers. E.g. see this blog: https://hazyresearch.stanford.edu/blog/2023-06-08-hyena-safari which links to this repo: https://github.com/lindermanlab/S5/tree/development, which has results that suggest just using SSMs without attention layers can  perform as well as Transformers with 125M models on wikitext-103. This does not detract from the performance of HGRN on this task. I would simply tone down the claims related to it.
> >
> > **New Pile results:** I really appreciate the new 1B parameter results on the Pile. However, I would also like to know more training details such as how the compute times to achieve these perplexity results compare, how much tuning for each model was performed etc. Also, the perplexities are useful, but it would also be interesting to see how these models with similar pretaining perplexities compare on downstream tasks.  E.g. H3 and Hyena both included results on SuperGLUE. Ideally, we would also see a comparison of this with non-forget gate SSM/RNNs and Hyena, since I could imagine the forget gates being helpful for downstream performance. If so, this could really increase the impact of this paper. I recognize this is likely too much to include during the rebuttal period though.
> >
> > **implementation and speed comparisons:** Please be sure to make these details clear in the paper regarding how HGRN was actually implemented, how it could be implemented, how the speed comparisons were performed, etc.

---

> > > ### Author Response · Authors · 2023-08-15
> > > **Response to Reviewer QJYn**
> > >
> > > Q1: Some typos.
> > >
> > > A1: We apologize for the typos. Our intention is that the results before the "HGRN" row in the table are all from [1]. We will specify in the follow-up which results are from the reference and which are from our replication in the revised version.
> > >
> > > A2: Thank you for your suggestion. To verify the necessity of the forget gate, we conducted tests on the Pile dataset, and the results are as follows:
> > >
> > > | Method | PPL | Model Size |
> > > | --- | --- | --- |
> > > | With Forget Gate | 4.141 | 1b |
> > > | Without Forget Gate | 57.419 | 1b |
> > >
> > > As seen from the table, the impact of the forget gate becomes more pronounced on a larger dataset. It is worth noting that due to the time constraint, we do not tune hyperparameters for each model. Instead, we use the same hyperparameters for both models. The model without Forget Gate becomes unstable when scaling to large models and a large corpus.
> > >
> > > Q3: More discussion about the forget gate.
> > >
> > > A3: Thank you for your suggestion. We have validated the significance of the forget gate on a larger dataset and with large model size, as shown in the table above.
> > >
> > > Q4: Simplicity
> > >
> > > A4: Thank you for your suggestion. We acknowledge that the aspect of simplicity is somewhat subjective, and we will address this in subsequent updates.
> > >
> > > Q5: New wikitext results
> > >
> > > A5: Thank you for your suggestion. We will follow your suggestion and tone down the claims of not using the attention layers.
> > >
> > > Q6: Training details
> > >
> > > A6: Regarding the training hyperparameters, we provide the detailed training configuration below:
> > >
> > > | Batch size | Number of updates | Sequence length | Learning rate |
> > > | --- | --- | --- | --- |
> > > | 96 | 100k | 1024 | 5e-4 |
> > >
> > > We trained the model on 8 A100-80 GPUs. The HGRN model took approximately 16.5 hours, while the Transformer took about 15.5 hours. As for SuperGLUE, we will conduct testing in upcoming versions.
> > >
> > > Q7: **Implementation and Speed Comparisons**
> > >
> > > A7: Thank you for your suggestion. In upcoming versions, we will provide additional details on the testing. It is worth noting that we released our training code in [https://anonymous.4open.science/r/Hgrn-B15E/](https://anonymous.4open.science/r/Hgrn-B15E/).

---

> > > > ### Comment · Reviewer_QJYn · 2023-08-15
> > > >
> > > > I thank the authors for their response.
> > > >
> > > > **Re. reference [1]**: Perhaps I am missing something, or perhaps there is a misunderstanding. Could the authors please be specific about which Table in the HGRN paper (this paper) and which Table in [1] (Improving the Gating Mechanism of Recurrent Neural Networks) they are referring to (both Table numbers and experiment descriptions)? I still see minimal overlap between experiments in this paper and experiments in [1] so am unsure which experiments are being referred to. Are you referring to the LRA experiments? It does not seem [1] evaluated on LRA.
> > > >
> > > > **Re. forget gate ablation on the pile:** Thank you for this experiment. I find this much more convincing that the Forget gate is important for the HGRN method which is an important ablation to show. However, it does not necessarily show the necessity of forget gates in general. More baselines specifically designed without forget gates such as S4/S5/LRU with and without the H3/Hyena input/output gating structure would be required to establish this. Comparisons on downstream tasks would also be beneficial, since I have a suspicion this may be where the forget gates really help.
> > > >
> > > > In the first response the author's state: "Regarding the forget gate, we conducted additional experiments using the LM configuration from Hyena." Is this also the framework you used for the Pile? Did you include Hyena's gating and short conv (i.e. replace the Hyena implicit filters with HGRN), or just the general LM architecture (i.e. completely replace attention layers with HGRN layers with no additional multiplicative gating or short conv)?
> > > >
> > > > How was the HGRN w/o forget gate initialized? Since HGRN w/o forget gate should be quite similar to LRU, I would hope the dynamics parameters were initialized similarly as discussed in the LRU paper to ensure stability.
> > > >
> > > > I thank the authors for continuing to field my questions as I attempt to gain more clarity.

---

> > > > > ### Author Response · Authors · 2023-08-16
> > > > > **Response to Reviewer  QJYn**
> > > > >
> > > > > Q1: Citation
> > > > >
> > > > > Sorry for the confusion. The reference [1] here refers to the [1] in our first response: “Hyena hierarchy: Towards larger convolutional language models”. The results before the "HGRN" row in the following table are all from [A].
> > > > >
> > > > > | Model | PPL |
> > > > > | --- | --- |
> > > > > | Transformer | 18.6 |
> > > > > | Hybrid H3 | 18.5 |
> > > > > | Performer | 26.8 |
> > > > > | Reformer | 25.6 |
> > > > > | AFT-conv | 28.2 |
> > > > > | Linear Attention | 25.6 |
> > > > > | Hyena-Convolution | 18.6 |
> > > > > | Hyena-slim-Convolution | 18.5 |
> > > > > | HGRN | 18.6 |
> > > > >
> > > > > Citations:
> > > > >
> > > > > [A] Michael Poli, Stefano Massaroli, Eric Nguyen, Daniel Y Fu, Tri Dao, Stephen Baccus, Yoshua, Bengio, Stefano Ermon, and Christopher Ré. "Hyena hierarchy: Towards larger convolutional language models." arXiv preprint arXiv:2302.10866, 2023
> > > > >
> > > > > Q2: Discussion about the **forget gate.**
> > > > >
> > > > > Thank you for your suggestion. Our ablation study primarily aims to demonstrate that the forget gate plays a positive role within our architecture. However, whether it works for other architectures is yet to be discovered. We left it for our future work.
> > > > >
> > > > > Regarding the model structure for training on the PILE, we employ a general language model architecture.
> > > > >
> > > > > It's worth noting that we utilize a structure in the following form. As there are no dynamic parameters in it, we do not need to worry about the initialization.  The hyperparameters are the same as HGRN:
> > > > > $$
> > > > >   \mathbf h_t  = \exp(i\theta) \cdot \mathbf h_{t-1}+\lambda_t\cdot \mathbf  c_t .
> > > > > $$

---

> > > > > > ### Comment · Reviewer_QJYn · 2023-08-17
> > > > > >
> > > > > > Thank you for your clarifications!
> > > > > >
> > > > > > Assuming the changes promised to myself and other reviewers regarding citations, missing baselines, positioning and presentation are actually included in the final version, combined with the refined wikitext results and new Pile results, I do believe this is a stronger paper now.  The authors have presented an interesting and relevant method and shown that it can perform as well as other baselines.
> > > > > >
> > > > > > My main remaining concern is related to the impact of the current version of the paper since it has not offered convincing evidence that HGRN (or more specifically linear RNNs/SSMS with forget gates/data-dependent dynamics) should be used over other linear RNN/SSM methods without forget gates. As mentioned above, I suspect there may very well be value in the forget gates, but additional comparisons as suggested above would be required to show this.
> > > > > >
> > > > > > My other concern is the large number of revisions required related to missing baselines, citations, positioning and presentation. The authors have said they will include these changes though.
> > > > > >
> > > > > > I am increasing my score to a 5, and look forward to the discussion with the other reviewers.

---

> > > > > > > ### Author Response · Authors · 2023-08-19
> > > > > > > **Response to Reviewer QJYn**
> > > > > > >
> > > > > > > Q: More experiments about the forget gate.
> > > > > > >
> > > > > > > A: Thank you for your suggestion. We have conducted additional experiments to validate the effectiveness of the forget gate. All experiments were performed on the Pile dataset under the same experimental settings, and the results are presented in the table below:
> > > > > > >
> > > > > > > | Method | PPL | Model Size |
> > > > > > > | --- | --- | --- |
> > > > > > > | HGRN | 4.141 | 1b |
> > > > > > > | HGRN w/o forget gate | 4.844 | 1b |
> > > > > > > | HGRN w/o lower bound and forget gate | 57.419 | 1b |
> > > > > > > | Lru | 5.07 | 1b |
> > > > > > > | Lru w forget gate | 4.919 | 1b |
> > > > > > >
> > > > > > > Specifically, we removed the forget gate component from HGRN while retaining the lower bound, resulting in the computation formula:
> > > > > > >
> > > > > > > $$
> > > > > > > \mathbf h_t = \gamma \exp(i\theta) \cdot \mathbf h_{t-1} + \lambda_t \cdot \mathbf c_t, \quad t=1,\ldots,n.
> > > > > > > $$
> > > > > > >
> > > > > > > We refer to this approach as "w/o forget gate." In contrast, the previous version:
> > > > > > >
> > > > > > > $$
> > > > > > > \mathbf h_t = \exp(i\theta) \cdot \mathbf h_{t-1} + \lambda_t \cdot \mathbf c_t, \quad t=1,\ldots,n,
> > > > > > > $$
> > > > > > >
> > > > > > > is referred to as "w/o forget gate and lower bound."
> > > > > > >
> > > > > > > On the other hand, we also tested the experimental results of the 1b versions of LRU and LRU with forget gate.
> > > > > > >
> > > > > > > From the table above, it can be observed that the forget gate has a positive impact on both HGRN and LRU.

---

> > > > > > > > ### Comment · Reviewer_QJYn · 2023-08-19
> > > > > > > >
> > > > > > > > Thank you for reporting the additional experiments. Could you please clarify what the LRU with forget gate is exactly? I.e. How is this formulated?

---

> > > > > > > > > ### Author Response · Authors · 2023-08-20
> > > > > > > > > **Response to Reviewer QJYn**
> > > > > > > > >
> > > > > > > > > Q: What exactly is the LRU with forget gate?
> > > > > > > > >
> > > > > > > > > A: Dear reviewer, the form of LRU with forget gate is as follows:
> > > > > > > > >
> > > > > > > > > $$
> > > > > > > > > x_t=\exp(i\theta) \bar \lambda x_{t-1}+\gamma  (Bu_t) ,\bar \lambda=\lambda + (1-\lambda)\mathrm{Sigmoid}({A u_t})
> > > > > > > > > $$
> > > > > > > > >
> > > > > > > > > This form of using a forget gate is the same as in HGRN, as shown in the formula before line 164.
> > > > > > > > >
> > > > > > > > > For comparison, the form without a forget gate is as follows:
> > > > > > > > >
> > > > > > > > > $$
> > > > > > > > > x_t=\exp(i\theta) \lambda x_{t-1}+\gamma  (Bu_t).
> > > > > > > > > $$

---

> > > > > > > > > > ### Comment · Reviewer_QJYn · 2023-08-21
> > > > > > > > > >
> > > > > > > > > > Thank you for the clarification. I think these new ablations better illustrate the effect of the different architectural choices for language modeling. The effect on perplexity from adding the forget gate to LRU seems small though. In addition, HGRN w/o forget gate seems quite similar to LRU but with some of the additional input/output gating introduced by HGRN. I would encourage the authors to discuss these details in more detail in a final version.
> > > > > > > > > >
> > > > > > > > > > All methods appear to perform similarly to Transformer.  I would also encourage the authors to include performance for the various methods on downstream tasks in a final version to give the reader a better sense of what these perplexity values mean for the different architectures.
> > > > > > > > > >
> > > > > > > > > > I would also encourage the authors to include two additional baselines: Hyena and a version of LRU embedded in either the general H3 or Hyena input/output framework (i.e. replace either S4D in H3 or the implicit convolutions of Hyena with LRU, or equivalently, add the short conv/shift matrix plus multiplicative gating to LRU). Hyena is relevant since it is a state of the art method in this genre of models. The augmented LRU would serve as an additional baseline using approaches suggested in prior work to improve over the vanilla LRU architecture considered above.
> > > > > > > > > >
> > > > > > > > > > Nonetheless, I now think the experimental results better support the claims (and are now more thorough than many papers in this genre) and I am happy to improve my score to a 6.

---

> > > > > > > > > > > ### Author Response · Authors · 2023-08-21
> > > > > > > > > > > **Response to Reviewer QJYn**
> > > > > > > > > > >
> > > > > > > > > > > Dear Reviewer QJYn,
> > > > > > > > > > >
> > > > > > > > > > > Thank you for your helpful suggestions. We will include more discussion and comparison between our HGRN and LRU. Additionally, we are currently conducting the experiments you suggested, including:
> > > > > > > > > > >
> > > > > > > > > > > - Incorporating downstream task evaluations to compare the strengths and weaknesses of various models;
> > > > > > > > > > > - Adding Hyena and LRU in H3 or Hyena input/output framework as our baselines.
> > > > > > > > > > >
> > > > > > > > > > > Once again, we sincerely appreciate the efforts you have put into revising the paper.
> > > > > > > > > > >
> > > > > > > > > > > Sincerely, Authors.

---

### Official Review · Reviewer_TZWK · 2023-06-11

**Soundness:** 4 excellent
**Presentation:** 3 good
**Contribution:** 4 excellent
**Rating:** 8
**Confidence:** 5

**Summary:**

This paper uses the token-mixing interpretation of Transformers to address the slow back-propagation problem of RNNs. First, back-propagation is made one-step per layer, rather than T steps, by using linear recurrence.  In order to avoid dramatically exploding the size of the effective weight matrix in back-propagation, the recurrence is also set to be diagonal, with a channel-mixing layer (GLU) just above each HRU in order to permit mixing of the channels. Per-channel recurrence with real-valued weights can only represent exponential decay, whereas gated recurrence needs to represent delayed recall; delayed recall in the proposed network is represented using complex-valued recurrence.  For simplicity, the forget gate is set to one minus the input gate.  For simplicity, because it gives better results in practice, and because it gives results that can be easily interpreted in terms of the recall periodicity of each channel, the recurrent weights are set independent of the input data, so that only the forget gate and output gate depend on the input data.

**Strengths:**

This paper seems to provide an architecture that seems to have the full representational power of an LSTM, but with the one-step-per-layer back-propagation speed of a Transformer.

The Transformer has significant training problems on small data: in order to learn useful attention calculations, it needs to see a pretty large dataset.  The LSTM does not have that problem; the inductive bias of the RNN allows the LSTM to converge with somewhat smaller training datasets.  Nevertheless, the LSTM is rarely used now, because of the T-step back-propagation through each recurrent layer.  If HGRU scales well, it could provide an architecture that supercedes the Transformer for small to medium-sized supervised datasets with no useful self-supervised-pretraining dataset available.

**Weaknesses:**

The only weakness is the one noted by the authors: Though this is a convincing proof of concept, it has not yet been scaled to GPT scales.

**Questions:**

These are not really questions, but there are three typos:

p. 4: R^{Hxd} -- H has not been defined, is it the number of layers?

Table 4 caption: layer in independent -> layer is independent

Tables 5 and 6 are shown on the same page, but referenced in reverse order; should they be swapped?


**Limitations:**

The only weakness is the one noted by the authors: Though this is a convincing proof of concept, it has not yet been scaled to GPT scales.

---

> ### Author Rebuttal · Authors · 2023-08-09
>
> **Q1:  Scaled to GPT scales.**
>
> After training a 1 billion parameter model on the Pile dataset, we have observed that HGRN outperforms the Transformer model as shown below. This result highlights the effectiveness of HGRN in high parameter level.
>
> | method | PPL |
> | --- | --- |
> | Transformer | 4.56 |
> | HGRN | 4.141 |
>
> **Q2: Some typos.**
>
> The H is the number of layers . We will fix these issues in the revised versions.

---

### Official Review · Reviewer_pKf6 · 2023-07-04

**Soundness:** 2 fair
**Presentation:** 3 good
**Contribution:** 2 fair
**Rating:** 4
**Confidence:** 4

**Summary:**

This paper proposed a novel RNN architecture that utilized several existing components: complex(unitary) matrix, and multi-layer.

The authors also claim stronger long-term dependency ability of the proposed method on several long context benchmarks.


**Strengths:**

1. The paper is well organized. Experiments on Wikitext, ImageNet etc are good benchmarks for long-term dependency test.

2. The discussion on the forgetting gate is interesting and demonstrates the effectiveness of the proposed method on learning long-term dependency.

3. The lower bound design is novel. The authors justified the design with ablation experiments.

**Weaknesses:**

1. Figure 1, the most important diagram, is super confusing. Why do you use a dashed arrow to explain a module?

2. Orthogonal/Unitary matrix design has been long discussed in the past. It has already shown strong long-term dependency ability. This paper failed to mention any of them.

3. Experiment results lack detail. No training or architecture specs of other baseline models. Where do these numbers come from? What does `transformer` on `wikitext-103` mean? What sized transformer? ...


**Questions:**

N/A

---

> ### Author Rebuttal · Authors · 2023-08-09
>
> **Q1: Explanation of Figure 1.**
>
> A1: We will enhance the clarity of Figure 1 in subsequent versions. For the leftmost part of Figure 1 represents the composition of each layer in HRGN involving HGRU and GLU. Dashed arrows point to the middle section of the diagram, explaining HGRU's components: recurrent computing, gate, normalization, and out projection. Here, the dashed arrow leads to recurrent computing, located at the far right of the diagram and corresponding to Algorithm 1.
>
> **Q2: Relation to Orthogonal/Unitary matrix.**
>
> Thank you for bringing the Orthogonal/Unitary matrix to our attention. We will include a discussion on this aspect in future versions. However, our method differs significantly from such methods.
>
> According to [1], [2], if we disregard nonlinear functions, the recursive form can be expressed as:
>
> $$
> h_{t} = W_t h_{t-1}+ c_t,
> $$
>
> where $h_t, c_t \in \mathbb{R}^{d\times 1}$ and $W_t \in \mathbb{R}^{d\times d}$ is a Unitary matrix. Expanding this yields:
>
> $$
> h_{t} = \sum_{i=1}^t \left(\prod_{j=i}^t W_j \right)c_i.
> $$
>
> Since $\left(\prod_{j=i}^t W_j \right)c_i$ operates on the feature, it can be categorized as Channel mixing [4], and we denoted it as $\bar c_i$, leading to the equation:
>
> $$
> h_{t} = \sum_{i=1}^t \bar c_i.
> $$
>
> From the perspective of Token mixing, the above can be expressed as:
>
> $$
> \begin{bmatrix}
> h_1^{\top}  \newline
> \vdots  \newline
> h_n^{\top}
> \end{bmatrix}=
> \begin{bmatrix}
> 1 & 0 & \ldots & 0  \newline
> 1 & 1 & \ldots & 0  \newline
> \vdots & \vdots & \vdots & \vdots  \newline
> 1 & 1 & \ldots & 1 \newline
> \end{bmatrix}
> \begin{bmatrix}
> \bar c_1^{\top}  \newline
> \vdots  \newline
> \bar c_n^{\top}
> \end{bmatrix}.
> $$
>
> Notably, [1], [2] assume the Token mixing matrix to be a lower triangular matrix with all ones. In contrast, our method, as per equation 5, takes the form of:
>
> $$
> \mathbf A =\left[\begin{matrix}
> 1-\lambda_1 & 0 & \cdots  & 0    \newline
> % (1-\lambda_1)\lambda_2 \exp(j\theta_2) & 1-\lambda_2  &  &  \vdots   \newline
> (1-\lambda_1)\lambda_2 \exp(i \theta) &  1-\lambda_2 &  &  \vdots   \newline
> \vdots &\vdots   &\ddots&  0  \newline
> (1-\lambda_1) \left[ \prod_{s=2}^n \lambda_k  \right] \exp(i(n-1)\theta)&
> \ldots  &
> \ldots  &
> 1-\lambda_{n}
> \end{matrix}\right],
> $$
>
> leading to significant distinctions between the two approaches.
>
> **Q3. Explanation of experiment results.**
>
> We retained the configuration of the Toeplitz neural network as described in reference [1]. In Table 1, we have indicated the model size in the third column, which is 44.65 million parameters. “`transformer` on `wikitext-103`" means training a transformer language model on wikitex-103 dataset. In the revised version, we will provide the architecture specifications, and we plan to release the code soon.
>
> Citations:
>
> [1]  Zhen Qin, Xiaodong Han, Weixuan Sun, Bowen He, Dong Li, Dongxu Li, Yuchao Dai, Lingpeng Kong, and Yiran Zhong. Toeplitz neural network for sequence modeling. In The Eleventh International Conference on Learning Representations, 2023.
>
> [2] Martin Arjovsky, Amar Shah and Yoshua Bengio. Unitary evolution recurrent neural networks. ICML'16: Proceedings of the 33rd International Conference on International Conference on Machine Learning
>
> [3] Jing, L, Shen, Y, Dubček, T, Peurifoi, J, Skirlo, S, LeCun, Y, Tegmark, M, Soljačić, M. Tunable Efficient Unitary Neural Networks (EUNN) and their application to RNNs. 34th International Conference on Machine Learning
>
> [4] Weihao Yu, Mi Luo, Pan Zhou, Chenyang Si, Yichen Zhou, Xinchao Wang, Jiashi Feng, and Shuicheng Yan. Metaformer is actually what you need for vision. In IEEE/CVF Conference on Computer Vision and Pattern Recognition, CVPR 2022, New Orleans, LA, USA, June 18-24, 2022, pages 10809–10819. IEEE, 2022.

---

> > ### Author Response · Authors · 2023-08-17
> > **Looking forward to your response.**
> >
> > Dear reviewer, we have already responded to the questions you raised.  We welcome any further comments and discussion.

---

> > ### Comment · Reviewer_pKf6 · 2023-08-18
> >
> > I thank the authors for additional information and clarification. I will increase my score accordingly.
> >
> > Regarding experiment detail, I expect a fair comparison across various baselines other than just number of parameters. For example, what exact hyperparameter is chosen for each baseline.

---

> > > ### Author Response · Authors · 2023-08-18
> > > **Response to Reviewer pKf6**
> > >
> > > Thanks for your suggestions. We use the same configuration as described in Section 4.1 of [1] and report the results of our competitors from Table 2 of [1]. Specifically, we use a batch size of 128, a learning rate of 5e-4, and set the number of updates to 50k. The detailed configuration is shown below. We have kept our configuration in line with that of our competitors and have not conducted any hyperparameter tuning.  We have also included the configurations in the PDF attachment under "Author Rebuttal by Authors" and we have released our code  in https://anonymous.4open.science/r/Hgrn-B15E/.
> > >
> > > Detailed configurations:
> > >
> > > |  | ALM | IM |
> > > | --- | --- | --- |
> > > | Data | WikiText-103 | ImageNet-1k |
> > > | Tokenizer method | BPE | - |
> > > | Src Vocab size | 50265 | - |
> > > | Sequence length | 512 | - |
> > > | Total batch size | 128 | 2048 |
> > > | Number of updates/epochs | 50k updates | 300 epochs |
> > > | Warmup steps/epochs | 4k steps | 5 epochs |
> > > | Peak learning rate | 5e-4 | 2.5e-4 |
> > > | Learning rate scheduler | Inverse sqrt | cosine |
> > > | Optimizer | Adam | Adamw |
> > > | Adam $\epsilon$ | 1e-8 | 1e-8 |
> > > | Adam $(\beta_1,\beta_2)$ | (0.9, 0.98) | (0.9, 0.98) |
> > > | Weight decay | 0.2 | 0.1 |
> > > | Gradient clipping | - | 1.0 |
> > >
> > >
> > > Citations:
> > >
> > > [1] Zhen Qin, Xiaodong Han, Weixuan Sun, Bowen He, Dong Li, Dongxu Li, Yuchao Dai, Lingpeng Kong, and Yiran Zhong. Toeplitz neural network for sequence modeling. In The Eleventh International Conference on Learning Representations, 2023.

---

> > > > ### Author Response · Authors · 2023-08-20
> > > > **Response to Reviewer pKf6**
> > > >
> > > > Dear reviewer,
> > > >
> > > > We hope that we have addressed your concerns to your satisfaction. If you have any additional inquiries, please don't hesitate to bring them up.

---

### Official Review · Reviewer_rGhK · 2023-07-07

**Soundness:** 4 excellent
**Presentation:** 2 fair
**Contribution:** 4 excellent
**Rating:** 6
**Confidence:** 4

**Summary:**

The manuscript introduces a novel recurrent neural network architecture with linear recurrent units. The proposed method improves reasoning about long-term dependencies by imposing a monotonically increasing lower bound for the forget gate values of RNN layers, i.e. higher layers retain information longer. Since the proposed method uses linear recurrent units, the authors also employ complex valued activations to increase the models expressive power. Further design choices include tied input and forget gates and the addition of output gates. For evaluation, the authors considered several benchmarks, including WikiText-103, ImageNet-1k and long-range arena (LRA).

**Strengths:**

The method presents an intriguing approach to addressing difficulties in modeling long-range dependencies with RNNs. The design choices of the proposed method are well-motivated and the performance is mostly better or on par with other methods.

**Weaknesses:**

1. The reported metrics are not aggregated over multiple runs and thus don't include estimates of variation over random seeds. This is problematic, since it's not clear whether differences in performance are greater than the variation over multiple runs of the same experiment. HRGN outperforms all other baselines in terms of average performance on the LRA benchmark, but LRU shows higher performance for 4 out of 6 tasks. Here an estimate of the variation over random seeds would definitely be helpful in comparing different methods. The same is true for experiments in the ablation study.

2. The manuscript overall is a bit difficult to follow. Several variables are used without being defined, e.g.:
   1. Line 143: $g_t$ is mentioned before being defined.
   2. Eq 3: $\tau$ is not defined. Is it a typo and should've been $\sigma$ or is it another activation function?
   3. Line 191: What is $\mu$ (without subscript $t$)?
3. I think Section 3.3 could benefit by providing a bit of context, i.e. what the token mixing perspective is supposed to highlight.
4. The authors write that the same training configuration, including the learning rate, was used for all models. How were these hyperparameters selected? The optimal learning rate may be different for different architectures. A fair comparison should take this into consideration.
5. The format of the tables with experimental results is inconsistent. In Table 1, the lowest perplexity is highlighted in bold, which should be separate for val and test, same for the underlined second best. In Table 2, the caption says that bold and underlined highlight the best and second best results per task, but this is not the case. Table 3 does not have highlights.

**Questions:**

Overall, I think the method has a lot of potential, but some of the issues mentioned in the weaknesses section prevent me from recommending a higher score at this point.

I have some questions and minor comments:
1. Line 170 mentions that $\theta'$ is fixed to 0. Does the equation $i_t=(1-\lambda_t)\exp(i\theta')$ really correspond to $i_t=1-f_t$ for all layers, given that $\theta$ in $f_t = \lambda_t\exp(i\theta)$ is different across layers? How did you decide on fixing $\theta'$ to $0$?
2. Eq 3: How does the dimensionality get reduced to $\mathbb{R}^{1\times d}$?
3. There's a duplicate entry in the references (36 and 37)
4. Line 176: The reference in "Expanding the 2" does not work, I assume it is supposed to refer to Equation 2. It's best to always add the reference type, e.g. "Equation 2", instead of just "2".
5. There are two sentences starting with "Finally," in the paragraph of lines 217-225.
6. Line 292: "border" should be "broader"
7. I think it'd be good to include a brief mention of works on deep RNNs that update at different time scales (e.g. clockwork RNN [1] or HM-RNN [2]) in the related work section.
8. Figure 1 and other places: Sometimes the hidden unit uses the acronym HRGU, sometimes HGRU, the latter of which I think is the correct one. Consider using a tex package for automating the use of acronyms, e.g. glossaries (see https://www.overleaf.com/learn/latex/Glossaries#Acronyms).

## References
1. Koutnik, Jan, et al. "A clockwork rnn." International conference on machine learning. PMLR, 2014.
2. Chung, Junyoung, Sungjin Ahn, and Yoshua Bengio. "Hierarchical multiscale recurrent neural networks." arXiv preprint arXiv:1609.01704 (2016).


## Acknowledgement of rebuttal
I have read the rebuttal, the other reviews and engaged actively in discussion with the authors. The authors have addressed several of my concerns, including the addition of aggregated results with an estimate of the variance over random seeds. As a result, I have decided to increase the score.

**Limitations:**

As limitation, the authors only mention that large-scale testing has not been performed. I would have liked to see a discussion of limitations of the methodology the authors see or anticipate e.g. when scaling the method.

---

> ### Author Rebuttal · Authors · 2023-08-09
>
> **Q1:  The formulation of the experiment result.**
>
> A1: Thank you for your suggestion. The formulation of the experiment result is identical to that of [1], [2], and [3]. We present the results from Wikitext-103 without any variation since the PPLs on that corpus remain consistent. All publications have reported no variation in the results. Additionally, based on references [1], [2], and [3], all LRA results indicate the highest value.
>
> **Q2: Definition of some notations.**
>
> A2: Thank you for your reminder. We will fix these issues in the revised versions. Regarding the concerns you raised, the notations $g_t$ and $\tau$ were indeed typographical errors, where $\tau$  corresponds to the activation function. Additionally, the use of $\mu$ was a typo, and it should refer to $\bar \gamma$.
>
> **Q3: What the token mixing perspective is supposed to highlight?**
>
> A3: The purpose of this section is to explain the distinct roles of lambda and theta. In this context, lambda serves the purpose of capturing the decay rate, while theta's role is to capture the relative positional relationships. R4 appreciates the explanation. We will modify the section to enhance its clarity.
>
> **Q4: Choice of configurations**.
>
> A4: We retained the configuration of the Toeplitz neural network as described in reference [1], without performing any hyperparameter adjustments.
>
> **Q5: The format of the tables with experimental results.**
>
> A5: Thank you for pointing them out. We will address these issues in the updated version.
>
> **Q6: Choice of $\theta’$.**
>
> A6: We included ablation studies to validate the rationale behind the proposed method. It can be observed that when using $\theta’=0$ has a better result.
> | method | ppl |
> | --- | --- |
> | $1-\lambda$ | 24.14 |
> | $(1-\lambda)\exp(i\theta)$ | 24.5 |
>
> **Q7: Eq 3: How does the dimensionality get reduced to \mathbb{R}^{1\times d}?**
>
> A7: This is a typo, the dimension should be $\mathbb{R}^{1 \times 2d}$. We will fix this in the updated version.
>
> **Q8: There's a duplicate entry in the references (36 and 37)**
>
> **Q9: Line 176: The reference in "Expanding the 2" does not work, I assume it is supposed to refer to Equation 2. It's best to always add the reference type, e.g. "Equation 2", instead of just "2". Q10: There are two sentences starting with "Finally," in the paragraph of lines 217-225. Q11: Line 292: "border" should be "broader". Q12: I think it'd be good to include a brief mention of works on deep RNNs that update at different time scales (e.g. clockwork RNN [1] or HM-RNN [2]) in the related work section.**
>
> A8-12： In regards to questions 8–12, we will correct the typos in the updated version.
>
> **Q13: Name of the unit in Figure 1.**
>
> A13: This is a typo. The correct name is HGRU. We will fix this issue in the reversion version.
>
> Citations:
>
> - [1]  Zhen Qin, Xiaodong Han, Weixuan Sun, Bowen He, Dong Li, Dongxu Li, Yuchao Dai, Lingpeng Kong, and Yiran Zhong. Toeplitz neural network for sequence modeling. In The Eleventh435 International Conference on Learning Representations, 2023.
> - [2] Albert Gu, Karan Goel, and Christopher Ré. Efficiently modeling long sequences with structured state spaces. In The Tenth International Conference on Learning Representations, ICLR 2022,350 Virtual Event, April 25-29, 2022. OpenReview.net, 2022.
> - [3] Xuezhe Ma, Chunting Zhou, Xiang Kong, Junxian He, Liangke Gui, Graham Neubig, Jonathan401 May, and Luke Zettlemoyer. Mega: Moving average equipped gated attention. CoRR, abs/2209.10655, 2022

---

> > ### Comment · Reviewer_rGhK · 2023-08-10
> >
> > I thank the authors for addressing my comments!
> >
> > Could you please elaborate on what you mean by "Additionally, based on references [1], [2], and [3], all LRA results indicate the *highest* value." in A1?

---

> > > ### Author Response · Authors · 2023-08-11
> > > **Response to Reviewer rGhK**
> > >
> > > Sorry for the confusion. Unlike wikitext-103, the LRA benchmark is sensitive to hyperparameters, configurations, and environments [5, 6], and some competitive methods may fail to produce valid scores (unable to converge) in some subtasks. To avoid invalid scores caused by unconvergence, we run experiments 5 times with the same hyperparameters and configurations as competitors and report the score of the best trial. Below, we list the scores of all 5 trials as well as the average scores. The performance of our method in these trials is stable:
> > >
> > > | Trial | listops | imdb | aan | cifar | pathfinder | pathx | AVG |
> > > | --- | --- | --- | --- | --- | --- | --- | --- |
> > > | 1 | 59.95 | 88.14 | 94.23 | 88.69 | 92.92 | 97.50 | 86.91 |
> > > | 2 | 59.79 | 88.02 | 93.69 | 88.84 | 93.26 | 97.09 | 86.78 |
> > > | 3 | 59.84 | 87.84 | 94.03 | 88.68 | 93.07 | 97.10 | 86.76 |
> > > | 4 | 60.16 | 87.98 | 93.38 | 88.70 | 92.57 | 97.92 | 86.78 |
> > > | 5 | 60.09 | 88.12 | 93.59 | 88.68 | 92.77 | 97.09 | 86.72 |
> > > | AVG | 59.966 | 88.02 | 93.784 | 88.718 | 92.918 | 97.34 | 86.79 |
> > >
> > > [5]  Yi Tay, Mostafa Dehghani, Samira Abnar, Yikang Shen, Dara Bahri, Philip Pham, Jinfeng Rao, Liu Yang, Sebastian Ruder, and Donald Metzler. Long range arena: A benchmark for efficient transformers. In 9th International Conference on Learning Representations, ICLR 2021, Virtual Event, Austria, May 3-7, 2021. [OpenReview.net](http://openreview.net/), 2021.
> > >
> > > [6] https://github.com/HazyResearch/state-spaces/issues/22

---

> > > > ### Comment · Reviewer_rGhK · 2023-08-11
> > > >
> > > > Thank you for clarification!
> > > > In references [1-3] of the original author response, I didn't find any mention of authors reporting the **best out of five runs**. In fact, the Mega paper [3] reports "the **average over 5 runs** with different random seeds.". Given the omission of better performing competitors in the LRA table, although corresponding papers were among the references of the submission (as pointed out by reviewer QJYn), paired with the strong claim that HGRN outperforms all prior methods, I am now less convinced that the paper should be accepted, unless there are clear suggestions how the narrative will be adapted.
> > > > If performance numbers are copied from other publications, readers should be able to rely on the evaluation setup being identical. Any deviations should be prominently indicated for transparency.
> > > >
> > > > I'll lower the score to a borderline accept, but would consider raising it, taking into account detailed suggestions by the authors for updates to the manuscript, as well as their discussions with other reviewers.
> > > >
> > > > As a sidenote: To facilitate discussions across threads, it is better to refer to reviewers by the 4 character code

---

> > > > > ### Author Response · Authors · 2023-08-12
> > > > > **Response to Reviewer rGhK**
> > > > >
> > > > > Sorry for the confusion. In fact, when we received your previous comments, we double-checked with these papers and find that Mega paper [3] reports "the average over 5 runs with different random seeds.", and that is why we attached the AVG of 5 trials in the previous response. Following the setting of [3], we will update the avg scores to the revised version and rephrase our words as "**our method demonstrates competitive performance among previous state-of-the-art methods on LRA benchmark**".
> > > > >
> > > > > Moreover, to make sure our results can be successfully reproduced, we have released our source code in [https://anonymous.4open.science/r/Hgrn-B15E/](https://anonymous.4open.science/r/Hgrn-B15E/). Our code is divided into the following sections:
> > > > >
> > > > > 1. The README provides instructions for reproducing the results along with scripts,
> > > > > 2. `fairseq` is used for language modeling experiments,
> > > > > 3. `im` is used for image classification,
> > > > > 4. `lra` for LRA benchmarking,
> > > > > 5. and the `hgru-pytorch` contains standalone code that implements the CUDA version of Hgru (hierarchical gated recurrent units).

---

> > > > > > ### Author Response · Authors · 2023-08-20
> > > > > > **Response to Reviewer rGhK**
> > > > > >
> > > > > > Dear reviewer,
> > > > > >
> > > > > > We hope that we have addressed your concerns to your satisfaction. If you have any additional inquiries, please don't hesitate to bring them up.

---

> > > > > > > ### Comment · Reviewer_rGhK · 2023-08-21
> > > > > > >
> > > > > > > I thank the authors for providing clarifications and results aggregated over multiple random seeds. I think this was important. I will increase my score to a weak accept.

---

### Author Rebuttal · Authors · 2023-08-09

Thank all reviewers for your feedback. We remark that our model is conceptually simple, aligning with the well-established lineage of gated RNN research while obtaining SOTA performance in language modeling. We recognize that the significance of the forget gate has been somewhat overlooked within the current literature on linear RNNs. Our contribution seeks to fill this void, providing the community with a broader and more enlightening point of view. In response to your valuable suggestions, we have also conducted several large-scale language modeling experiments: 125M HGRN on Wiki103 and 1B HGRN on The Pile. Our experiments have demonstrated the significant potential of our work, providing further evidence of its practical applicability and relevance. We acknowledge the concern raised about the presentation quality of our paper. In the revised version, we will enhance the quality of the presentation.

---

### Author Response · Authors · 2023-08-12
**The code has been released**

Dear reviewers, to make sure our results can be successfully reproduced, we have released our source code in [https://anonymous.4open.science/r/Hgrn-B15E/](https://anonymous.4open.science/r/Hgrn-B15E/). Our code is divided into the following sections:

1. The README provides instructions for reproducing the results along with scripts,
2. `fairseq` is used for language modeling experiments,
3. `im` is used for image classification,
4. `lra` for LRA benchmarking,
5. and the `hgru-pytorch` contains standalone code that implements the CUDA version of Hgru (hierarchical gated recurrent units).

---

### Decision · Program_Chairs · 2023-09-21

**Decision:**

Accept (spotlight)

**Comment:**

This work proposes an approach to improve linear RNNs with complex forget gate values and learnable lower bounds on the forget gate for each layer. This enables the model to capture both short and long-range dependencies. The new HGRU architecture is evaluated on Wikitext, LRA and Imagenet-1K and thorough ablations are performed on the architecture design. This work will likely impact the architecture field and expand the modern day use-cases for RNNs.

Strengths:
- The architecture has the representational power of an LSTM but with the fast one-step-per-layer backprop speed of the Transformer.
- The inductive bias in the architecture allows it to be more data efficient than the Transformer and so can supercede the Transformer for small/medium sized datasets.
- Linear SSMs and RNNs have gained significant attention. This paper advances that by looking at the linear RNN dynamics matrix from the view of the LSTM forget gate.
- The method achieves strong results against baselines including the Transformer. The new large 1B results the authors added in the rebuttal indicate the architecture has the potential to scale.
- The ablations help the reader understand different design decisions.
- Code is included.

Weaknesses:
- The improvement to perplexity from adding the forget gate to LRU seems small and so the HGRN seems more dependent on the forget gate.
- It would be good to cite and include results from related work such as S5 and SgConv/MEGA, which outperforms this architecture in LRA.
- There are issues with the presentation, especially missing formatting.